# SOUNDREACTOR: FRAME-LEVEL ONLINE VIDEO-TO-AUDIO GENERATION

## ABSTRACT

Prevailing Video-to-Audio (V2A) generation models operate offline, assuming an entire video sequence or chunks of frames are available beforehand. This critically limits their use in interactive applications such as live content creation and emerging generative world models. To address this gap, we introduce the novel task of frame-level online V2A generation, where a model autoregressively generates audio from video without access to future video frames. Furthermore, we propose SoundReactor, which, to the best of our knowledge, is the first simple yet effective framework explicitly tailored for this task. Our design enforces end-to-end causality and targets low per-frame latency with audio-visual synchronization. Our model's backbone is a decoder-only causal transformer over continuous audio latents. For vision conditioning, it leverages grid (patch) features extracted from the smallest variant of the DINOv2 vision encoder, which are aggregated into a single token per frame to maintain end-to-end causality and efficiency. The model is trained through a diffusion pre-training followed by consistency fine-tuning to accelerate the diffusion head decoding. On a benchmark of diverse gameplay videos from AAA titles, our model successfully generates semantically and temporally aligned, high-quality full-band stereo audio, validated by both objective and human evaluations, at low per-frame token-level latency (26.6ms for the head NFE=1, 30.3ms for NFE=4 with 30FPS, 480p videos using a single H100.). Demo samples are available at https://anonymous-sr-submission.github.io/.

## 1 INTRODUCTION

Video-to-Audio (V2A) generation, the task of generating semantically and temporally aligned, high-quality audio from videos, is an extensive research area driven not only by its fundamental multimodal alignment challenges but also by its wide range of real-world applications, such as film production, video content creation, and game development (Luo et al., 2023; Du et al., 2023; Mei et al., 2024b; Zhang et al., 2024; Wang et al., 2024b;a; Viertola et al., 2025; Cheng et al., 2025; Zhong et al., 2025; Liu et al., 2025). Recent advances have led to models capable of generating high-fidelity audio with strong audio-visual synchronization (Cheng et al., 2025; Liu et al., 2025). However, the prevailing paradigm operates in an offline setting, assuming that an entire video sequence or chunks of frames are available in advance. This fundamentally limits their use in frame-level online applications.

In addition to limiting online applications in live content creation, this offline constraint also becomes critical in the context of world models, which generate video sequences in an online manner from frame-level conditions (Schmidhuber, 1990; Ha & Schmidhuber, 2018; Bruce et al., 2024; Baldassarre et al., 2025; Ball et al., 2025). Recent models, such as Genie 3 (Ball et al., 2025), can generate video frames in real-time (Ball et al., 2025; Microsoft, 2025; He et al., 2025; Li et al., 2025a), yet they remain silent, operating solely in the pixel domain. Since audio and vision are tightly coupled in both real and simulated worlds, extending world models to the audio-visual domain could broaden their applicability to entertainment, education, and interactive world simulation for robotics (Chen et al., 2020a; Liu et al., 2024b; NVIDIA et al., 2025), open-ended learning (Team, 2021), and agent training (Chen et al., 2022a; Team, 2024). To bring sound to these worlds requires moving beyond the offline assumption toward a new paradigm of frame-level conditioned online general audio generation.

To address this, we introduce the novel task of **frame-level online** V2A generation, where **a model autoregressively generates audio corresponding to a video stream without accessing future**

Figure 1: Our scope is frame-level online video-to-audio (V2A) generation task, **where future video frames are not available in advance**. This contrasts with conventional offline V2A task, where an entire video sequence or chunk of frames is available in advance.

**frames** (See Figure 1). The online setting imposes additional unique challenges beyond those inherent to the offline setting[1]: **1). End-to-end causality**: The entire autoregressive (AR) framework, including the vision encoding, should be causal, as future video frames are inaccessible beforehand, **2). Low per-frame latency**: models should generate audio at low per-frame latency for real-time interactive applications. Notably, even state-of-the-art V2A systems that use AR models (e.g., V-AURA (Viertola et al., 2025)) do not enforce end-to-end causality on their vision encoders (We refer to Section 4.2 and Appendix A for more details.).

To tackle these challenges, we introduce SoundReactor, the first simple yet effective framework tailored for the frame-level online V2A task on full-band stereo audio. Our design is driven by three essential properties: end-to-end causality, low per-frame latency, and high-quality audio generation with semantic and temporal audio-visual alignment. Our framework consists of an image encoder, a waveform encoder, and a causal, decoder-only multimodal Transformer (Vaswani et al., 2017) with a diffusion head (Li et al., 2024) (See Figure 2). For the image encoder, we introduce a novel conditioning scheme based on a pretrained DINOv2 encoder (Oquab et al., 2024) (Figure 2 (a)). We extract spatial grid (patch) features from each frame, a process that is inherently causal. The availability of its lightweight variants (e.g., 21M parameters) contributes to low per-frame latency for vision encoding. Moreover, we further augment these grid features with temporal differences, providing both semantic and temporal information to the generative model backbone. For continuous audio latents, we utilize a Variational Autoencoder (VAE) (Kingma & Welling, 2014) (Figure 2 (b)). We choose continuous latents for two reasons. First, for complex data such as full-band stereo audio, continuous representations tend to yield superior reconstruction quality over discrete counterparts at the same temporal downsampling rate (Lan et al., 2024), which is crucial for the final generation quality [2]. Second, by representing each frame with a single audio latent, it simplifies AR modeling compared to predicting multiple code indices per frame as typically required by residual vector quantization (RVQ)-based tokenizers (Kumar et al., 2023; Défossez et al., 2023; Li et al., 2025b). Finally, to reduce inference latency, we accelerate the diffusion head with Easy Consistency Tuning (ECT) (Geng et al., 2025b).

We demonstrate the efficacy of SoundReactor on the OGameData dataset (Che et al., 2025), a large-scale dataset of diverse AAA gameplay videos, motivated by the field of world models. Through our experiments, we show that our model generates semantically and temporally synchronized, high-quality, full-band stereo audio under the end-to-end causal constraint. This performance is validated by both objective and human evaluations and achieved with low per-frame token-level latency (26.6ms with the head NFE=1 and 30.3ms with NFE=4), measured on 480p, 30FPS videos using a single H100 GPU.

Our main contributions are:

- We introduce the frame-level online V2A generation task, a new paradigm as a key step toward interactive multimodal applications.

---

[1]e.g., generating high-quality audio with strong semantic and temporal audio-visual alignment.

[2]While the reconstruction quality of discrete tokenizers can be improved by increasing their bitrate, training high-bitrate tokenizers is a non-trivial task.

- We propose SoundReactor, which is, to our knowledge, the first framework explicitly tailored to this task, featuring a novel visual conditioning scheme and continuous audio latents for AR generation with an accelerated diffusion head.
- We demonstrate that SoundReactor achieves notable V2A generation performance under the end-to-end causal constraint while maintaining low per-frame token-level latency.

## 2 PRELIMINARIES

### 2.1 DIFFUSION MODELS

Let $p_{\text{data}}$ denote the data distribution. In Diffusion Models (DMs) (Ho et al., 2020), a clean sample $\mathbf{x}^0 \sim p_{\text{data}}$ is generated through an iterative denoising process, beginning from a Gaussian prior $p_T$. When the forward diffusion process is defined by $d\mathbf{x}^t = \sqrt{2t}\, d\mathbf{w}_t$, initialized by $\mathbf{x}^0$, where $\mathbf{w}_t$ is the standard Wiener process in forward-time (Karras et al., 2022; 2024), the deterministic counterpart of the denoising process, called the Probability Flow Ordinary Differential Equation (PF-ODE) (Song et al., 2021), is formulated as

$$\frac{d\mathbf{x}^t}{dt} = -t\nabla \log p_t(\mathbf{x}^t). \tag{1}$$

Typically, DMs are trained by minimizing a Denoising Score Matching (DSM) objective (Vincent, 2011; Song et al., 2021) expressed as:

$$\mathbb{E}_{\mathbf{x}^0, t, \boldsymbol{\epsilon}}[\|\mathbf{x}^0 - D_{\boldsymbol{\theta}}(\mathbf{x}^t, t)\|_2^2], \tag{2}$$

where $\boldsymbol{\epsilon} \sim \mathcal{N}(\mathbf{0}, \mathbf{I})$, $\mathbf{x}^t = \mathbf{x}^0 + t\,\boldsymbol{\epsilon}$, and $D_{\boldsymbol{\theta}}(\mathbf{x}^t, t)$ is a neural approximation of a denoiser function $\mathbb{E}_{p_{0|t}(\mathbf{x}^0|\mathbf{x}^t)}[\mathbf{x}^0|\mathbf{x}^t] = \mathbf{x}^t + t^2 \nabla \log p_t(\mathbf{x}^t)$ (Efron, 2011).

### 2.2 AUTOREGRESSIVE MODELING WITHOUT DISCRETE TOKENIZATION

AR V2A generation models have been typically implemented on discrete tokens (Iashin & Rahtu, 2021; Sheffer & Adi, 2023; Mei et al., 2024b; Du et al., 2023; Viertola et al., 2025) (A broader literature review of V2A generation models is provided in Appendix A.1.) In contrast, several recent works decouple discrete tokenization from AR modeling (Li et al., 2024; Tschannen et al., 2025b;a). Specifically, MAR (Li et al., 2024) models per-token conditional distributions via a DM. Our framework builds on MAR, and we decode one token (here, a continuous-valued latent[3]) at a time in chronological order. We refer to this sampling variant as MAR throughout[4].

Below, we outline both sampling and training procedures of MAR. Let $\mathbf{z}_i \in \mathbb{R}^{c_{\mathbf{z}}}$ denote an output vector by an AR model $F_{\boldsymbol{\phi}}$ at position $i$. Decoding in MAR proceeds in two stages. First, the conditioning vector $\mathbf{z}_i$ is estimated by previous tokens: $\mathbf{z}_i = F_{\boldsymbol{\phi}}(\texttt{<BOS>}, \mathbf{z}_1, \ldots, \mathbf{z}_{i-1})$, where $\texttt{<BOS>}$ is a start token to generate $\mathbf{z}_1$. Second, a vector $\mathbf{x}_i^0$ is sampled from the probability $q(\mathbf{x}_i^0 \mid \mathbf{z}_i)$ via an iterative denoising procedure (e.g., Eq. 1) using a trained conditional denoiser $D_{\boldsymbol{\theta}}(\mathbf{x}_i^t, t, \mathbf{z}_i)$.

For training, the AR model $F_{\boldsymbol{\phi}}$ and the diffusion head $D_{\boldsymbol{\theta}}$ are jointly trained by minimizing a diffusion loss (Ho et al., 2020). In this work, we employ the following DSM objective based on EDM (Karras et al., 2022; 2024):

$$\mathcal{L}(\boldsymbol{\theta}, \boldsymbol{\phi}) = \mathbb{E}_{\mathbf{x}^0, t, \boldsymbol{\epsilon}}\left[ \sum_{i=1}^{n} \left\| \mathbf{x}_i^0 - D_{\boldsymbol{\theta}}(\mathbf{x}_i^t, t, \mathbf{z}_i) \right\|_2^2 \right]. \tag{3}$$

### 2.3 EASY CONSISTENCY TUNING

Despite their success, DMs are slow at sampling, often requiring tens to hundreds of steps to generate a single sample. This limitation is particularly severe for our frame-level online V2A setting, where the AR model with the diffusion head incurs non-negligible per-frame latency. To address the slow generation of DMs, substantial community efforts have been made to reduce the number of sampling steps while preserving quality (Song et al., 2023; Song & Dhariwal, 2024; Lu & Song, 2025; Geng

---

[3]Unless otherwise stated, we denote "token" as a continuous-valued latent.

[4]The original MAR predicts tokens autoregressively with various protocols.

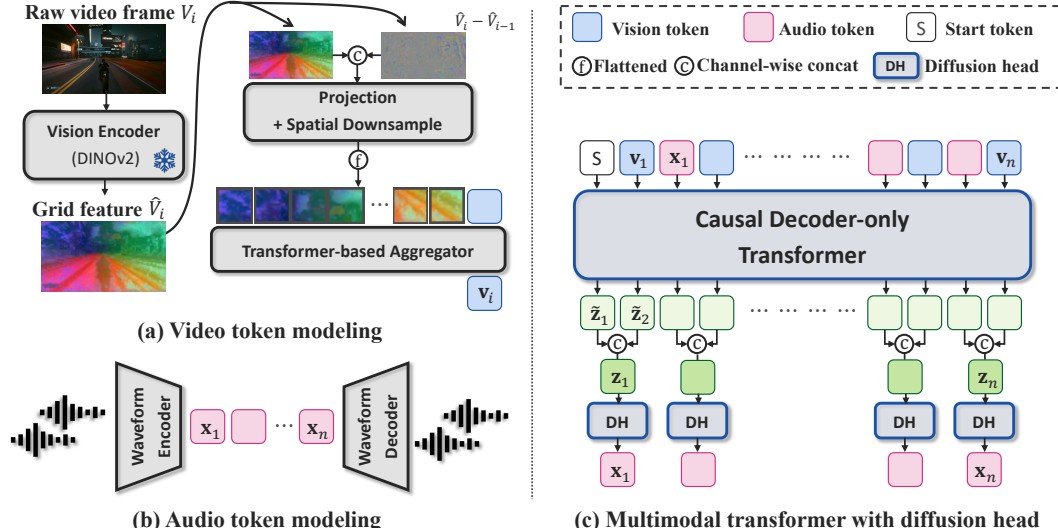

**Figure 2:** Overview of SoundReactor. Our framework has three components: (a) Video token modeling, (b) Audio token modeling, and (c) Multimodal AR transformer with diffusion head.

et al., 2025b; Kim et al., 2024; Zhou et al., 2025; Geng et al., 2025a; Saito et al., 2025; Novack et al., 2025b;a; Bai et al., 2024).

Within the family of Consistency Models (CMs) (Song et al., 2023; Song & Dhariwal, 2024; Lu & Song, 2025), ECT (Geng et al., 2025b) smoothly interpolates from DMs to CMs by progressively tightening the consistency condition, thereby bootstrapping a pretrained DM into a CM. It further shows that initializing a CM with a pretrained DM yields superior performance under a smaller training budget.

CMs are grounded in the PF-ODE (Eq. 1) and learn a consistency function $G_{\boldsymbol{\theta}}(\mathbf{x}^t, t)$ that maps a noisy sample $\mathbf{x}^t$ back to its clean data $\mathbf{x}^0$ as $G_{\boldsymbol{\theta}}(\mathbf{x}^t, t) = \mathbf{x}^0$. In ECT, $G_{\boldsymbol{\theta}}(\mathbf{x}^t, t)$ is trained by minimizing the following objective:

$$\mathbb{E}_{\mathbf{x}^0, t, r, \boldsymbol{\epsilon}} \left[ w(t) d\big( G_{\boldsymbol{\theta}}(\mathbf{x}^t, t), G_{\mathrm{sg}(\boldsymbol{\theta})}(\mathbf{x}^r, r) \big) \right], \tag{4}$$

where $t > r > 0$, $\mathbf{x}^t = \mathbf{x}^0 + t\,\boldsymbol{\epsilon}$, $\mathbf{x}^r = \mathbf{x}^0 + r\,\boldsymbol{\epsilon}$, $w(t)$ is a weighting function, $\mathrm{sg}(\boldsymbol{\theta})$ indicates an exponential moving average (EMA) of the past values of $\boldsymbol{\theta}$, and $d$ is a metric function. During training, $\Delta t = t - r$ start large (DMs) and is gradually annealed toward $0$ (CMs). Note that $\mathbf{x}^t$ and $\mathbf{x}^r$ are using shared noise $\boldsymbol{\epsilon}$.

## 3 SOUNDREACTOR

### 3.1 PROBLEM SETUP: FRAME-LEVEL ONLINE VIDEO-TO-AUDIO GENERATION

Motivated by Section 1, we aim to train a frame-level online V2A generator. Given sequences of audio tokens $\{\mathbf{x}_1, \ldots, \mathbf{x}_n\}$ and video tokens $\{\mathbf{v}_1, \ldots, \mathbf{v}_n\}$, where $\mathbf{x}_i \in \mathbb{R}^{c_\mathbf{x}}$ and $\mathbf{v}_i \in \mathbb{R}^{c_\mathbf{v}}$ denote frame-indexed vectors for $i = 1, \ldots, n$, the frame-level online V2A task is to model the conditional distribution of audio given video frames under an end-to-end causal constraint:

$$p(\mathbf{x}_{1:n} \mid \mathbf{v}_{1:n}) := \prod_{i=1}^{n} p\big( \mathbf{x}_i \mid \mathbf{x}_{<i}, \mathbf{v}_{\leq i} \big). \tag{5}$$

We aim to train a simple yet effective generator that models $p(\mathbf{x}_i \mid \mathbf{x}_{<i}, \mathbf{v}_{\leq i})$, where the video tokens are provided in an online manner, **i.e., an entire pipeline cannot access future video frames in advance**, as illustrated in Figure 1. Throughout the paper, our models are based on continuous-valued tokens for both audio and video. In this work, we assume a frame-aligned pairing between tokens (i.e., $\mathbf{x}_i$ and $\mathbf{v}_i$ refer to the same time frame.).

## 3.2 MODELING FRAMEWORK

We first outline an overview of SoundReactor, then describe each component and the motivation behind its design.

**Model overview** Figure 2 summarizes SoundReactor. Our framework has three components: (a) Video token modeling: a pretrained vision encoder projects each video frame $V_i \in \mathbb{R}^{H \times W \times 3}$ to a single aggregated token $\mathbf{v}_i$; (b) Audio token modeling: a VAE encodes full-band stereo waveforms into a sequence of continuous-valued tokens $\mathbf{x}_i$; and (c) Multimodal AR transformer with diffusion head: frame-aligned, interleaved audio–visual tokens are fed into a causal decoder-only transformer trained with a next-token prediction by an EDM2-style DSM loss.

**Video token modeling** Figure 2 (a) depicts our video token modeling. We adopt a pretrained DINOv2 vision encoder (Oquab et al., 2024) to extract grid (patch) features $\hat{V}_i \in \mathbb{R}^{H' \times W' \times c_{\text{dinov2}}}$ per video frame. To inject temporal cues, each frame's features are then concatenated with their temporal difference from the previous frame ($\hat{V}_i - \hat{V}_{i-1}$) along the channel dimension. These features pass through a projection layer with a 2D convolutional downsampler and are then flattened. We then prepend a learnable aggregation token and feed this sequence to a shallow transformer aggregator, yielding a single token $\mathbf{v}_i$ per frame. We use grid features instead of the encoder's [CLS]-token because, in our preliminary analysis, [CLS]-token lacked temporal cues needed for audio-visual synchronization (described in more detail in Appendix B.1). We choose DINOv2 for its availability of a lightweight variant (21 M parameters) for encoding efficiency and for its rich semantic representations (Ranzinger et al., 2024). Note that leveraging pretrained DINOv2 and its grid features from vision encoders as video conditioning for V2A generation remains underexplored.

**Audio token modeling** Figure 2 (b) provides our audio token modeling. Following the VAEs from Stable Audio (SA) series (Evans et al., 2025; 2024), we compress stereo 48 kHz waveforms into a sequence of audio tokens $\mathbf{x}_i$. Since the original SA-VAE is not trained at 48 kHz, we train it from scratch. Using continuous-valued tokens brings two benefits for AR audio generation. First, without discrete tokenization (Kumar et al., 2023; Défossez et al., 2023; Li et al., 2025b), reconstruction quality is typically higher at the same temporal downsampling rate (Lan et al., 2024). Second, AR modelings can be simplified because waveform VAEs represent each time frame with a single latent, allowing AR models to sample one token per frame. In contrast, especially in the full-band general audio settings, discrete tokenizers commonly rely on RVQ, which typically requires AR models to predict multiple code indices per frame (Copet et al., 2023; Défossez et al., 2024).

**Multimodal transformer with diffusion head** In Figure 2 (c), we present the framework of the multimodal causal decoder-only transformer with the diffusion head. We feed frame-aligned, interleaved audio–visual continuous-valued tokens into the model. The AR backbone follows LLaMA-style design (Touvron et al., 2023a;b), applying pre-normalization using RMSNorm (Zhang & Sennrich, 2019), SwiGLU activation (Shazeer, 2020), and rotary positional embeddings (RoPE) (Su et al., 2024). The diffusion head largely follows the original architecture in MAR. We follow the EDM2 (Karras et al., 2024) training framework for a DSM loss; accordingly, we add a simple one-layer MLP to parameterize an uncertainty function, which we describe in Section 3.3. To obtain conditional token $\mathbf{z}_i$, which is fed to the diffusion head, we first conduct channel-wise concatenation $\mathbf{z}_i = \texttt{Concat}(\tilde{\mathbf{z}}_{2i-1}, \tilde{\mathbf{z}}_{2i})$, where $\tilde{\mathbf{z}}_{2i-1}$ and $\tilde{\mathbf{z}}_{2i}$ are the outputs of transformer depicted in Figure 2 (c). By following MAR, we then project $\mathbf{z}_i$ to the head's channel dimension.

## 3.3 TRAINING FRAMEWORK

**Stage 1: Diffusion pretraining** We first pretrain the model with a next-token prediction by minimizing a DSM objective under the EDM2 framework. The overall training loss is formulated as:

$$\mathcal{L}_{\text{Stage 1}}(\boldsymbol{\theta}, \boldsymbol{\phi}) = \mathbb{E}_{\mathbf{x}^0, t, \boldsymbol{\epsilon}} \left[ \lambda(t) e^{-u_{\boldsymbol{\theta}}(t)} \sum_{i=1}^{n} \left\| \mathbf{x}_i^0 - D_{\boldsymbol{\theta}}(\mathbf{x}_i^t, t, \mathbf{z}_i) \right\|_2^2 + u_{\boldsymbol{\theta}}(t) \right], \tag{6}$$

where $\mathbf{z}_i = F_{\boldsymbol{\phi}}(\mathbf{v}_{\leq i}, \mathbf{x}_{<i}^0)$, a loss weighting $\lambda(t) = (t^2 + \sigma_{\text{data}}^2)/(t \cdot \sigma_{\text{data}})^2$, $\ln t \sim \mathcal{N}(P_{\text{mean}}, P_{\text{std}})$, $\sigma_{\text{data}}$ is the standard deviation of the training data, and $u_{\boldsymbol{\theta}}(t)$ is a one-layer MLP-based continuous uncertainty function quantifying the uncertainty for the denoising objective at noise level $t$ (Karras et al., 2024). We provide the training algorithm of Stage 1 in Algorithm 1.

**Stage 2: Consistency fine-tuning via ECT**    We further apply ECT to accelerate the diffusion head toward low per-frame latency. We minimize the following objective:

$$\mathcal{L}_{\text{Stage 2}}(\boldsymbol{\theta}, \boldsymbol{\phi}) = \mathbb{E}_{\mathbf{x}^0, t, r, \boldsymbol{\epsilon}} \left[ w(t) \sum_{i=1}^{n} d\big(G_{(\boldsymbol{\theta}, \boldsymbol{\phi})}(\mathbf{x}_i^t), G_{\text{sg}(\boldsymbol{\theta}, \boldsymbol{\phi})}(\mathbf{x}_i^r)\big) \right], \tag{7}$$

where $G_{(\boldsymbol{\theta}, \boldsymbol{\phi})}(\mathbf{x}_i^t) \triangleq G_{(\boldsymbol{\theta}, \boldsymbol{\phi})}(\mathbf{x}_i^t, t, \mathbf{v}_{\leq i}, \mathbf{x}_{<i}^0)$ is the entire network (including diffusion head $D_{\boldsymbol{\theta}}$ and transformer $F_{\boldsymbol{\phi}}$.). We initialize a model weight with the Stage 1 model. During training, $\Delta t = t - r$ is gradually annealed as $\Delta t \to 0$ by following a mapping function $m(r|t, \text{Iters})$ along with the number of training iterations Iters. We use $m(r|t, \text{Iters})$ formulated as:

$$m(r|t, \text{Iters}) = \left( 1 - \frac{1}{q^{\lceil \text{Iters}/s \rceil}} k\sigma(-bt) \right) t, \tag{8}$$

where $\sigma(\cdot)$ is the sigmoid function, $q$, $s$, $k$, and $b$ are hyperparameters that control the annealing schedule. Note that we finetune both the transformer and the diffusion head on Stage 2 (See an ablation on this in Appendix C.1.). We provide the training algorithm of Stage 2 in Algorithm 2.

### 3.4 SAMPLING FRAMEWORK

At inference time, following the MAR decoding paradigm, we operate a two-stage procedure. First, we obtain $\mathbf{z}_i$ by channel-wise concatenation $\texttt{Concat}(\tilde{\mathbf{z}}_{2i-1}, \tilde{\mathbf{z}}_{2i})$, where $\tilde{\mathbf{z}}_{2i-1}$ and $\tilde{\mathbf{z}}_{2i}$ are outputs of the transformer. Second, the diffusion head samples $\mathbf{x}_i^0$ conditioned on $\mathbf{z}_i$ through the reverse-diffusion sampling with the Stage 1 model or CM sampling with the Stage 2 model.

To get $\tilde{\mathbf{z}}_{2i}$ (a corresponding input token is a vision condition $\mathbf{v}_i$), we apply Classifier-Free Guidance (CFG) (Ho & Salimans, 2022) on the transformer's sampling as $\tilde{\mathbf{z}}_{2i} = \tilde{\mathbf{z}}_{2i}^{\text{null}} + \omega(\tilde{\mathbf{z}}_{2i}^{\text{cond}} - \tilde{\mathbf{z}}_{2i}^{\text{null}})$, where $\omega$ is a guidance scale, $\tilde{\mathbf{z}}_{2i}^{\text{cond}}$ is the transformer output given the visual condition $\mathbf{v}_i$, and $\tilde{\mathbf{z}}_{2i}^{\text{null}}$ is the output given a learnable null embedding, respectively. To get $\tilde{\mathbf{z}}_{2i}^{\text{null}}$, all the $\mathbf{v}_{\leq i}$ are replaced with the learnable null embedding. To enable CFG sampling, we randomly replace all the video tokens $\mathbf{v}_i (i = 1, \ldots, n)$ by the null embedding during training. We use KV-cache (Shazeer, 2019) for efficient inference. We provide the sampling procedure in Algorithm 3. Since our CFG sampling scheme is not identical to the original MAR, we describe the difference in Appendix B.3 with an ablation study in Appendix C.2.

## 4 EXPERIMENTS

### 4.1 DATASET

Following the context of world models, we evaluate SoundReactor on the OGameData250K (Che et al., 2025) dataset, a large collection of gameplay videos from diverse AAA titles. From the original 250,000 videos (14-16 seconds each), we collected 230K available samples and manually filtered out clips containing only background music. This results in a training set of 94K samples (around 400 hours) and a test set of $3,830$ samples. To ensure a balanced representation of each game, we create the train and test splits by maintaining a consistent clip ratio within each game title. All data is processed at 30 FPS with 480p resolution and 48 kHz stereo audio. During training, we randomly sample 8-second clips. Our primary evaluation is conducted on 8-second clips. We also test our models on longer sequence generation beyond the training context window on 16-second clips using zero-shot context-window extension techniques (Chen et al., 2023; Peng et al., 2024).

### 4.2 COMPARING METHODS

As discussed in the preceding sections, our scope is the frame-level online V2A task. To the best of our knowledge, no prior work explicitly addresses this setting. We therefore compare against offline AR V2A models, which are the most analogous to our setup. Specifically, we choose V-AURA (Viertola et al., 2025), as it achieves state-of-the-art performance in audio quality, audio-visual semantic alignment, and temporal synchronization among recent AR V2A models (Sheffer & Adi, 2023; Du et al., 2023; Mei et al., 2024b; Viertola et al., 2025).

**V-AURA**    V-AURA employs a LLaMA-style architecture with Synchformer (Iashin et al., 2024) as a video encoder and DAC (Kumar et al., 2023) as a discrete tokenizer. We train V-AURA on

Table 1: FAD and MMD on OGameData. Center is average over Left and Right channels. Diff is their difference. We retrained V-AURA. Note that V-AURA's video encoder extracts chunk-level features using non-causal self-attention across frames, so it is not strictly compatible with frame-level online V2A task. Bold and underlined scores indicate best and second-best results, respectively.

| Model | Channels/ Sample rate | FAD↓ | | | | | | | | MMD↓ | | | | | | | |
| | | OpenL3 | | | | LAION-CLAP | | | | OpenL3 | | | | LAION-CLAP | | | |
| | | Center | Diff | Left | Right | Center | Diff | Left | Right | Center | Diff | Left | Right | Center | Diff | Left | Right |
| **Baselines** | | | | | | | | | | | | | | | | | |
| VAE-Reconstruction | 2/48kHz | 33.0 | 20.2 | 34.9 | 32.2 | 0.058 | 0.009 | 0.063 | 0.055 | 60.4 | 28.7 | 67.1 | 60.1 | 0.301 | 0.518 | 0.321 | 0.260 |
| V-AURA† (Viertola et al., 2025) | 1/44.1kHz | 68.3 | – | 69.5 | 69.2 | 0.287 | – | 0.284 | 0.285 | 110 | – | 113 | 113 | 1.84 | – | 1.81 | 1.80 |
| **Proposed methods** | | | | | | | | | | | | | | | | | |
| SoundReactor-Diffusion | 2/48kHz | 36.8 | 25.7 | 40.1 | 37.8 | 0.139 | 0.257 | 0.159 | 0.155 | 67.6 | 37.2 | **77.6** | 71.5 | 0.702 | 1.52 | 0.828 | 0.787 |
| SoundReactor-ECT (NFE=1) | 2/48kHz | 38.2 | 29.1 | 43.0 | 39.7 | 0.129 | 0.299 | 0.147 | 0.146 | 78.8 | 54.2 | 95.2 | 85.9 | 0.649 | 1.81 | 0.735 | 0.729 |
| SoundReactor-ECT (NFE=2) | 2/48kHz | 35.1 | 24.8 | 39.5 | 36.1 | 0.105 | 0.250 | 0.121 | 0.115 | 70.0 | 43.1 | 85.0 | 74.9 | 0.509 | 1.47 | 0.595 | 0.549 |
| SoundReactor-ECT (NFE=4) | 2/48kHz | **33.9** | **22.6** | **37.7** | **34.3** | **0.090** | **0.219** | **0.107** | **0.100** | **64.6** | 35.4 | 77.1 | **67.5** | **0.451** | **1.29** | **0.527** | **0.473** |

Table 2: FSAD, KL$_{\text{PaSST}}$, IB-Score, and DeSync on OGameData. Bold score indicates the best results. We retrained V-AURA.

| Model | FSAD↓ | KL$_{\text{PaSST}}$↓ | IB-Score↑ | DeSync↓ |
| --- | --- | --- | --- | --- |
| **Baselines** | | | | |
| VAE-Reconstruction | 0.00796 | 0.643 | 0.260 | 0.943 |
| V-AURA† (Viertola et al., 2025) | 0.113 | 1.96 | 0.223 | **0.982** |
| **Proposed methods** | | | | |
| SoundReactor-Diffusion | 0.0219 | 1.61 | **0.292** | 1.06 |
| SoundReactor-ECT (NFE=1) | 0.0254 | 1.61 | 0.274 | 1.02 |
| SoundReactor-ECT (NFE=2) | 0.0206 | 1.60 | 0.277 | 1.01 |
| SoundReactor-ECT (NFE=4) | **0.0180** | **1.57** | 0.285 | 1.04 |

Table 3: Subjective evaluation on audio quality (Oval), semantic alignment (AV-Sem), temporal alignment (AV-Temp), and Stereo panning correctness (Stereo) with $95\%$ Confidence Interval.

| Model | Oval ↑ | AV-Sem ↑ | AV-Temp ↑ | Stereo ↑ |
| --- | --- | --- | --- | --- |
| Ground-truth | 83.6±2.44 | 82.2±2.71 | 83.4±2.52 | 81.2±2.76 |
| V-AURA† | 42.2±3.93 | 42.5±4.15 | 50.5±4.07 | 43.2±3.82 |
| **Ours-ECT (NFE=1)** | 61.9±3.07 | 64.5±3.20 | 58.4±3.55 | 60.5±3.08 |
| **Ours-ECT (NFE=4)** | **64.9±3.01** | **65.2±3.19** | **64.3±3.53** | **65.3±3.00** |

offline V2A setups with OGameData250K by following the original training setups suggested by the authors. We retrain the transformer part. Crucially, although V-AURA's backbone is the AR transformer, its video encoder, Synchformer (Iashin et al., 2024), extracts chunk-level features using non-causal self-attention across frames (Iashin et al., 2024; Bertasius et al., 2021; Patrick et al., 2021) (i.e., Synchformer extracts chunks of video frames, and its video features contain future information), so it is not strictly compatible with the frame-level online V2A task[5].

**SoundReactor** We evaluate two main variants of our model: SoundReactor-Diffusion (after Stage 1) and SoundReactor-ECT (after Stage 2). We also employ the reconstruction from our VAE as a reference. Unless otherwise specified, SoundReactor-Diffusion is sampled with a 30-step deterministic Heun solver (a Number of Function Evaluation (NFE)=59). We set $\omega = 3.0$ to get $\tilde{\mathbf{z}}_{2i}$ (See Section 3.4). We investigate the influence of $\omega$ in Appendix C.2.

### 4.3 EVALUATION METRICS

**Audio quality** To evaluate audio quality, we use Frechet Audio Distance (FAD) (Gui et al., 2024), Maximum Mean Discrepancy (MMD) (Chung et al., 2025; Jayasumana et al., 2024), and Kullback-Leibler divergence based on PaSST (Koutini et al., 2022) (KL$_{\text{PaSST}}$). For FAD and MMD, following prior work Evans et al. (2024; 2025); Gui et al. (2024), we use OpenL3 (Cramer et al., 2019) and LAION-CLAP (Wu* et al., 2023) as audio feature extractors. Since our model is trained on stereo signals, we compute the metrics on four channel configurations: the left channel (Left), the right channel (Right), their average (Center), and their difference (Diff) (Steinmetz et al., 2021). We compute KL$_{\text{PaSST}}$ only on the Center. Furthermore, as a complementary metric to evaluate stereo quality, we also use Frechet Stereo Audio Distance (FSAD) (Sun et al., 2025), which is based on StereoCRW (Chen et al., 2022b) features.

**Audio-visual alignment** To evaluate audio-visual semantic and temporal alignment, we utilize ImageBind score (IB-Score) (Girdhar et al., 2023) and DeSync (Viertola et al., 2025) by following common practices (Cheng et al., 2025; Viertola et al., 2025). To compute DeSync, we take two crops (first 4.8 sec. and last 4.8 sec.) from the video and compute the average over the results (Cheng et al., 2025). We compute these two metrics only on the Center.

**Listening test** To complement our objective metrics, we conduct a subjective listening test following the MUSHRA style (ITU-R BS.1534-3). We compare three methods: V-AURA, SoundReactor-ECT (NFE=1), and SoundReactor-ECT (NFE=4). From a pool of 50 video clips, each of 17 human evaluators is presented with 10 randomly selected samples per method in addition to the ground truth as a hidden reference. Participants are asked to rate the generated audio based on four criteria on a

---

[5]We note that their work targets the offline V2A task.

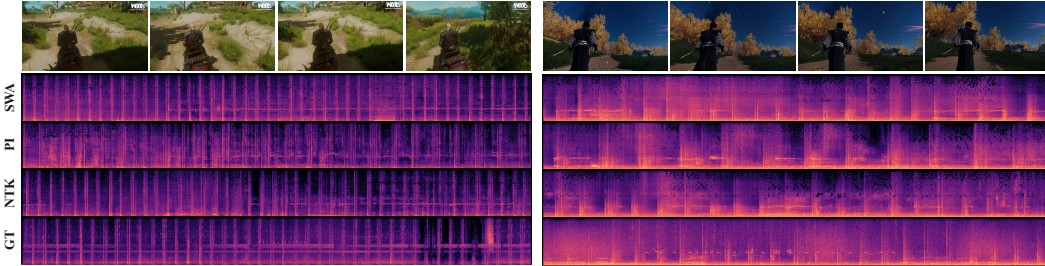

Figure 3: Spectrograms of long-seq. generation (twice the training window) with SoundReactor-ECT (NFE=4) comparing using Sliding Window Attention (SWA), Position Interpolation (PI), NTK-aware Interpolation (NTK) and the ground-truth (GT). PI results in a slower cadence for periodic sound.

scale of 0 to 100: overall audio quality (Oval), audio-visual semantic alignment (AV-Sem), temporal alignment (AV-Temp), and stereo quality and panning accuracy (Stereo). The evaluation interface is shown in Figure 9.

## 4.4  RESULTS

**Effectiveness on frame-level online V2A generation**  The results of the quantitative and subjective evaluations are presented in Tables 1, 2, and 3. For these evaluations, V-AURA's monaural signal is duplicated for the left and right channels. Consequently, its Diff is omitted as the difference becomes zero. These tables show that our methods outperform V-AURA on all metrics except for DeSync, indicating the effectiveness of SoundReactor on the frame-level online V2A generation task. The results of FSAD in Table 2

Table 4: Performance on 16-sec. long-sequence generation, **evaluated on first and last** 8**-sec. segments, independently**. All methods generate the full 16-sec. sequence continuously, except for Baseline that generates two separate 8-sec.

| Method | First 8 sec. | | Last 8 sec. | |
|---|---|---|---|---|
| | $FAD_{CLAP}\downarrow$ | $MMD_{CLAP}\downarrow$ | $FAD_{CLAP}\downarrow$ | $MMD_{CLAP}\downarrow$ |
| Baseline | | | | |
| Ours-ECT | 0.116 | 0.466 | 0.115 | 0.477 |
| Long-seq. gen. | | | | |
| Ours w/. SWA | **0.111** | **0.461** | 0.141 | 0.678 |
| Ours w/. PI | 0.154 | 0.734 | 0.147 | 0.703 |
| Ours w/. NTK | 0.116 | 0.477 | **0.111** | **0.459** |

and Stereo in Table 3 demonstrate that our model can generate stereo audio with panning accuracy with visual sound-event locations. We refer the generated audio samples to the demo page[6]. Comparing the performance of SoundReactor-Diffusion and ECTs in 1 to 3, the ECTs show comparable performance with the Diffusion even at NFE=1 on the head, and surpass it at NFE=4. While SoundReactor-Diffusion uses NFE=59, SoundReactor-ECTs require only 1~4 NFEs for the diffusion head, yielding $14.8\times$ fewer NFEs at NFE=4 and $59\times$ fewer at NFE=1, with a small performance degradation. We measure per-frame latency in wall-clock time in a later paragraph. As a supplementary evaluation, we benchmark our framework against various offline V2A approaches on the VGGSound dataset (Chen et al., 2020b) in Appendix C.4.

**Generation beyond training context window**  We test our model on longer sequence generation beyond the training context window using zero-shot context-window extension techniques. Specifically, we use 16-second test set samples ($1,206$ samples in total), which is twice the training context window. We apply and compare two common zero-shot RoPE scaling techniques to extend the context window: Position Interpolation (PI) (Chen et al., 2023) and NTK-aware interpolation (NTK) (Peng et al., 2024) (See Appendix B.4.) and Sliding Window Attention

Table 5: Overall performance on 16-sec. sequences. Features are extracted from 8-sec. sliding windows with a 1-sec. hop size. These features are then aggregated via mean-pooling to compute metrics.

| Method | $FAD_{CLAP}\downarrow$ | $MMD_{CLAP}\downarrow$ |
|---|---|---|
| Ours w/. SWA | 0.108 | 0.475 |
| Ours w/. PI | 0.138 | 0.668 |
| Ours w/. NTK | **0.0992** | **0.412** |

(SWA)[7] (Beltagy et al., 2020; Jiang et al., 2023). All the methods are applied on SoundReactor-ECT (NFE=4). Table 4 demonstrates that the NTK does not show significant performance degradation across either the first or last 8-second segment, despite extending the context window. In contrast, PI shows degradation throughout the entire sequence, while SWA degrades in the second half. This indicates that NTK-aware scaling effectively mitigates the distribution shift caused by long-sequence generation. In terms of overall quality, Table 5 shows that NTK and SWA are competitive, with

---

[6]https://anonymous-sr-submission.github.io/

[7]Note that context window is not extended on SWA.

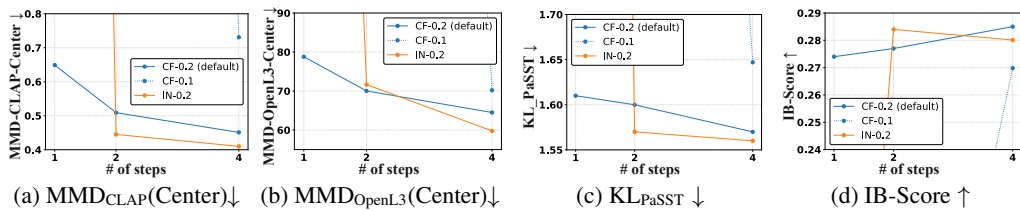

| (a) MMD$_{CLAP}$(Center)↓ | (b) MMD$_{OpenL3}$(Center)↓ | (c) KL$_{PaSST}$ ↓ | (d) IB-Score ↑ |

Figure 4: Ablation study on ECT. `CF` and `IN` indicate different mapping functions (see Appendix B.2). 0.1 and 0.2 denote dropout rates (Srivastava et al., 2014) during finetuning. CF−0.2 is our default.

PI performing worst. These results demonstrate that, by leveraging the NTK, our model achieves zero-shot context-window extrapolation without having the distribution shift. Given that interactive multimodal applications might require minute- to hour-scale generation, we leave exploring longer-horizon generation as future work. From qualitative observation, the spectrogram visualization in Figure 3 shows that PI slows periodic sounds (e.g., footsteps) and harms temporal synchronization, while NTK and SWA preserve timing (See Appendix B.4 for further discussion).

**Per-frame latency measurement** We measure two types of latency: wave-level and token-level latency. Wave-level latency is the elapsed time from feeding the previous audio token $x_{n-1}$ into the model until the output $x_n$ is incrementally decoded into a waveform, including the encoding of a raw video frame $V_n$. Token-level latency is identical to this but excludes the waveform decoding (Detailed in Appendix B.5). We perform this benchmark on a single H100 GPU with a batch size of one, using a single

Table 6: Mean and standard deviation of per-frame latencies over 50 clips. Lower than 33.3ms on wave-level latency is real-time operation.

| Model (Ours-ECT) | Token-level latency [ms]↓ | Wave-level latency [ms]↓ |
|---|---|---|
| NFE=1 | $26.6 \pm 2.93$ | $32.4 \pm 1.91$ |
| NFE=2 | $27.4 \pm 2.60$ | $34.7 \pm 2.13$ |
| NFE=4 | $30.3 \pm 2.41$ | $38.6 \pm 1.46$ |

CUDA stream. We provide the mean and standard deviation over the 50 clips (30FPS, 480p, 16 second) with the default our ECT at $\omega = 3$. As demonstrated in Table 6, while our model achieves remarkably low per-frame token-level latency, the overhead from incremental waveform decoding prevents the end-to-end wave-level latency from meeting the 33.3 ms target (real-time for 30FPS videos) except at NFE=1, which falls short of the generation quality of NFE=2 and 4. Addressing this bottleneck to enable real-time high-quality generation for practical applications remains an important direction for future work.

## 4.5 ABLATION STUDY

Here, we ablate two factors in ECT that crucially affect sample quality: i) the mapping function in Eq (8) and ii) the network dropout rate (Srivastava et al., 2014). We use two representative choices from Geng et al. (2025b), denoted `IN` and `CF` (detailed in Appendix B.2). Under `CF`, we test dropout rates of 0.1 and 0.2 (CF−0.2 is identical to our default ECT.). As shown in Figure 4, comparing CF−0.1 and CF−0.2 reveals that the dropout rate substantially impacts sample quality. This aligns with prior findings in Song & Dhariwal (2024); Geng et al. (2025b) and newly shows that tuning it remains significantly important even in MAR-style models. Comparing IN−0.2 and CF−0.2, both settings show comparable performance on NFE=2 and 4. Notably, however, only CF−0.2 can generate high-quality samples with NFE=1. For a deeper understanding of our model, we provide extensive ablations on various aspects of both the generator (in Appendix C.1 and C.2) and vision conditioning (in Appendix C.3), which demonstrate that our design choices contribute to the overall performance.

## 5 CONCLUSION

In this work, we introduce the novel task of frame-level online V2A generation and propose SoundReactor, the first simple yet effective framework tailored for this setting. Our experiments demonstrate that SoundReactor achieves high-quality full-band stereo audio generation under the end-to-end causal constraint while maintaining low per-frame token-level latency. Extensive ablations provide key insights into the respective contributions of our design choices within our framework. We anticipate this work will motivate the research community on sound generative models to explore interactive multimodal applications, providing a foundational component for live content creation and sounding world models.

## ETHICS STATEMENT

SoundReactor poses a risk for generating harmful or inappropriate content, offensive sound effects, or even sound content that infringes on copyrights. To mitigate these risks, it is crucial to carefully curate the training data as a first step. Furthermore, addressing these risks involves implementing robust content filtering, moderation mechanisms to prevent the creation of unethical, harmful content.

## REPRODUCIBILITY STATEMENT

The source code is available at the submitted supplementary materials, and the checkpoints are available at https://drive.google.com/drive/folders/1AVfogi4nxKlVesDi-LP4migAuN8C3jp7?usp=sharing. Moreover, we outline our training and sampling procedures in Algorithm 1, Algorithm 2, and Algorithm 3, and detailed implementation instructions for result reproducibility can be found in Appendix B.1. After acceptance, we will open-source our codebase.

## THE USE OF LARGE LANGUAGE MODELS

We utilized LLMs for academic proofreading and finding related work. We also used them for coding assistance and visualization of experimental results. However, all research ideas were developed solely by the authors.

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

CONTENTS

## A  RELATED WORK

### A.1  VIDEO-TO-AUDIO GENERATION

Offline video-to-audio (V2A) generation has been explored using both autoregressive (AR) and non-autoregressive (Non-AR) models.

**AR models.**  AR models for V2A have focused on generating discrete audio tokens. One line of work operates on VQ-VAE (van den Oord et al., 2017) latents derived from Mel-spectrograms, which requires a separate vocoder for waveform synthesis. This category includes SpecVQGAN (Iashin & Rahtu, 2021), which uses a transformer to generate code indices from video RGB and optical flow; Im2Wav (Sheffer & Adi, 2023), which adopts a dual-transformer architecture conditioned on CLIP (Radford et al., 2021) features; and CondFoleyGen (Du et al., 2023), which uses ResNet (2+1)D-18 vision encoder (He et al., 2016). Another line of AR models utilizes discrete tokenizers based on raw waveforms using RVQ (Défossez et al., 2023; Kumar et al., 2023). In this paradigm, FoleyGen (Mei et al., 2024b) explores various vision encoders like CLIP and ImageBind (Girdhar et al., 2023), while V-AURA (Viertola et al., 2025) has achieved state-of-the-art performance in this category by introducing Synchformer (Iashin et al., 2024) video encoder, which enabled semantic and strong temporal alignment. The Synchformer's video encoder, a Motionformer (Patrick et al., 2021; Bertasius et al., 2021) relies on non-causal attention over video chunks. While suitable for the offline task, this makes V-AURA incompatible with our scope, the frame-level online V2A task. Our framework, SoundReactor, is categorized in this AR paradigm based on continuous audio latents and is applicable for both the offline and frame-level online V2A tasks.

**Non-AR models.**  Non-AR models have also been widely investigated. The first direction involves training models from scratch based on Diffusion Models (DMs) (Ho et al., 2020) and Flow-matching Models (FMs) (Liu et al., 2023b). For instance, Diff-Foley (Luo et al., 2023) uses a CAVP (Luo et al., 2023) vision encoder to train DMs on Mel-spectrogram latents, while MultiFoley (Chen et al., 2025) applies a similar approach to latents derived from raw waveforms projected by DAC-based VAE (Kumar et al., 2023). Models such as Frieren (Wang et al., 2024b) and MMAudio (Cheng et al., 2025) leverage FMs; Frieren uses CAVP, whereas MMAudio employs a multimodal diffusion transformer (Esser et al., 2024) with CLIP and Synchformer encoders. ThinkSound (Liu et al., 2025) builds upon a multimodal LLM framework that uses Chain-of-Thought reasoning, followed by the MMAudio-style audio generator. A second direction is training models from scratch based on discrete tokens using MaskGIT (Chang et al., 2022). VATT (Liu et al., 2024a) employs a two-stage pipeline where it first generates a text caption from the video and then uses MaskGIT to predict code indices of EnCodec (Défossez et al., 2023). MaskVAT (Pascual et al., 2024) applies MaskGIT to predict code indices of DAC conditioned on CLIP and SparseSync (Iashin et al., 2022) features. The third major trend is an adaptation of pre-trained text-to-audio models. Several methods steer AudioLDM (Liu et al., 2023a): ReWaS (Jeong et al., 2024) uses a ControlNet-based approach (Zhang et al., 2023) with an energy curve from the Synchformer; V2A-Mapper (Wang et al., 2024a) employs a lightweight network to map CLIP embeddings to CLAP embeddings (Wu* et al., 2023); Seeing & Hearing (Xing et al., 2024) conditions the generator using ImageBind as a multimodal aligner; and FoleyCrafter (Zhang et al., 2024) fine-tunes it using an adapter-based method with CLIP features and a learnable timestamp adapter. Other backbones are also adapted, such as SpecMaskGIT (Comunità et al., 2024) by SpecMaskFoley (Zhong et al., 2025) and Stable Audio Open (Evans et al., 2025) by CAFA (Benita et al., 2025), both using ControlNet with CLIP and Synchformer features.

### A.2  AR AUDIO GENERATION WITHOUT VECTOR QUANTIZATION

Our work builds upon the emerging paradigm of AR audio generation with continuous audio latents. Prior work in this area has focused on unimodal audio domains such as unconditional music generation (CAM (Pasini et al., 2024)) and speech synthesis (DiTAR (Jia et al., 2025), SALAD (Turetzky et al., 2024), MELLE (Meng et al., 2025)). Concurrently, CALM (Rouard et al., 2025) adopts a MAR-style framework (Li et al., 2024) for speech and music continuation, using a consistency model-based head. Our work is distinct from this prior and concurrent work in two key aspects. First, to our best knowledge, we are the first to tackle an audio-visual multimodal task within this paradigm, where

---

**Algorithm 1** SoundReactor-Diffusion's training

---

**Require:** Dataset $\mathcal{D}$
1: **repeat**
2:  Sample $(\{\mathbf{x}_1^0, \ldots, \mathbf{x}_n^0\}, \{\mathbf{v}_1, \ldots, \mathbf{v}_n\}) \sim \mathcal{D}, \boldsymbol{\epsilon}_i \sim \mathcal{N}(0, I), t \sim p(t)$     $\triangleright \ln t \sim \mathcal{N}(P_{\text{mean}}, P_{\text{std}})$
3:  $\{\mathbf{v}_1, \ldots, \mathbf{v}_n\} \leftarrow \varnothing$ with $p_{\text{uncond}}$               $\triangleright \varnothing$: null-embedding
4:  $\mathbf{x}_i^t \leftarrow \mathbf{x}_i^0 + t \, \boldsymbol{\epsilon}_i$
5:  $\mathbf{z}_i \leftarrow F_{\boldsymbol{\phi}}(\mathbf{v}_{\leq i}, \mathbf{x}_{<i}^0)$
6:  $\mathcal{L}_{\text{Stage1}}(\boldsymbol{\theta}, \boldsymbol{\phi}) \leftarrow \lambda(t) e^{-u_{\boldsymbol{\theta}}(t)} \sum_{i=1}^{n} \left\| \mathbf{x}_i^0 - D_{\boldsymbol{\theta}}(\mathbf{x}_i^t, t, \mathbf{z}_i) \right\|_2^2 + u_{\boldsymbol{\theta}}(t)$
7:  Update $(\boldsymbol{\theta}, \boldsymbol{\phi}) \leftarrow (\boldsymbol{\theta}, \boldsymbol{\phi}) - \eta \nabla_{(\boldsymbol{\theta}, \boldsymbol{\phi})} \mathcal{L}_{\text{Stage1}}$
8: **until** converged
9: **return** $(\boldsymbol{\theta}, \boldsymbol{\phi})$

---

**Algorithm 2** SoundReactor-ECT's training

---

**Require:** Dataset $\mathcal{D}$, pretrained parameters $(\tilde{\boldsymbol{\theta}}, \tilde{\boldsymbol{\phi}})$, mapping $m(r \mid t, \text{Iters})$, weighting function $w(t)$, metric function $d$,
1: **Init:** $(\boldsymbol{\theta}, \boldsymbol{\phi}) \leftarrow (\tilde{\boldsymbol{\theta}}, \tilde{\boldsymbol{\phi}}), \text{Iters} \leftarrow 0$
2: **repeat**
3:  Sample $(\{\mathbf{x}_1^0, \ldots, \mathbf{x}_n^0\}, \{\mathbf{v}_1, \ldots, \mathbf{v}_n\}) \sim \mathcal{D}$
4:  Sample $\boldsymbol{\epsilon}_i \sim \mathcal{N}(0, I), t \sim p(t), r \sim m(r \mid t, \text{Iters})$    $\triangleright \ln t \sim \mathcal{N}(P_{\text{mean}}, P_{\text{std}})$
5:  $\{\mathbf{v}_1, \ldots, \mathbf{v}_n\} \leftarrow \varnothing$ with $p_{\text{uncond}}$              $\triangleright \varnothing$: null-embedding
6:  $\mathbf{x}_i^t \leftarrow \mathbf{x}_i^0 + t \, \boldsymbol{\epsilon}_i, \quad \mathbf{x}_i^r \leftarrow \mathbf{x}_i^0 + r \, \boldsymbol{\epsilon}_i, \quad \Delta t \leftarrow t - r$
7:  $\mathcal{L}_{\text{Stage2}}(\boldsymbol{\theta}, \boldsymbol{\phi}) \leftarrow w(t) \sum_{i=1}^{n} d\big(G_{(\boldsymbol{\theta}, \boldsymbol{\phi})}(\mathbf{x}_i^t), G_{\text{sg}(\boldsymbol{\theta}, \boldsymbol{\phi})}(\mathbf{x}_i^r)\big)$   $\triangleright$ sg: stop-gradient
8:  $(\boldsymbol{\theta}, \boldsymbol{\phi}) \leftarrow (\boldsymbol{\theta}, \boldsymbol{\phi}) - \eta \nabla_{(\boldsymbol{\theta}, \boldsymbol{\phi})} \mathcal{L}_{\text{Stage2}}(\boldsymbol{\theta}, \boldsymbol{\phi})$
9:  $\text{sg}(\boldsymbol{\theta}, \boldsymbol{\phi}) \leftarrow \text{stopgrad}(\mu \, \text{sg}(\boldsymbol{\theta}, \boldsymbol{\phi}) + (1 - \mu)(\boldsymbol{\theta}, \boldsymbol{\phi}))$      $\triangleright \mu$: EMA rate
10:  $\text{Iters} \leftarrow \text{Iters} + 1$
11: **until** $\Delta t \rightarrow 0$
12: **return** $(\boldsymbol{\theta}, \boldsymbol{\phi})$

---

we introduce and solve the novel task of frame-level online V2A generation (see Section 1 and 3.1 for comparison to the offline setup.). Second, regarding diffusion head acceleration, our work is complementary to CALM: we investigate the ECT framework (Geng et al., 2025b), in contrast to their sCM-based from-scratch training (Lu & Song, 2025).

# B  EXPERIMENTAL DETAILS

## B.1  IMPLEMENTATION DETAILS

**Vision token modeling** We adopt a pretrained DINOv2 vision encoder to extract grid features per video frame. We use grid features instead of the encoder's `[CLS]`-token because it lacks the temporal cues necessary for audio-visual synchronization. Our preliminary analysis, shown in Figure 5, reveals that the cosine similarities between (A) adjacent video frames (30FPS) and (B) every other frame (15FPS) are significantly high[8]. We extract grid features from the final layer before the `[CLS]`-token head. We employ the `dinov2-vits14-reg`[9] (21 M parameters) for better per-frame encoding efficiency than larger backbones. Since raw grid features might be computationally expensive depending on a budget (e.g., training throughput and data storage), we first apply bilinear interpolation to halve both spatial dimensions. Second, we apply Principal Component Analysis (PCA) (Jolliffe, 2011) to reduce the hidden dimension from the original 384. We use 59, which preserves 70% of the cumulative explained variance (CEV) by default. These compressed features are precomputed before training. The adjacent frame subtraction is conducted on the precomputed grid features before being fed to the downstream projection layer described in Section 3.2 and Figure 2 (a). We use a 512-dimensional learnable aggregate token for the transformer aggregator. For the transformer

---

[8]The average cosine similarity over $3,830$ video clips in our test split is $0.99 \pm 0.0072$ for adjacent frames and $0.98 \pm 0.010$ for every other frame.

[9]https://github.com/facebookresearch/dinov2

---

**Algorithm 3** SoundReactor's frame-level online V2A sampling

---

**Require:** Video frames $\{V_i\}_{i=1}^n$ (Raw videos); vision encoding module $\mathcal{E}$; transformer $F_\phi$; diffusion head $D_\theta$; waveform decoder $\mathrm{Dec}_{\mathrm{VAE}}$; null embedding $\varnothing$; guidance scale $\omega$; head mode $\mathtt{mode} \in \{\mathtt{Diffusion}, \mathtt{ECT}\}$

1: $\mathbf{x}_0 \leftarrow$ `<BOS>`
2: **for** $i = 1 \dots n$ **do**
3:   $\tilde{\mathbf{z}}_{2i-1} \leftarrow F_\phi(\mathbf{x}_{i-1})$
4:   $\mathbf{v}_i \leftarrow \mathcal{E}(V_i)$
5:   $[\tilde{\mathbf{z}}_{2i}^{\mathrm{cond}}, \tilde{\mathbf{z}}_{2i}^{\mathrm{null}}] \leftarrow F_\phi([\mathbf{v}_i, \varnothing])$       $\triangleright$ Batch process for CFG paths
6:   $\tilde{\mathbf{z}}_{2i} \leftarrow \tilde{\mathbf{z}}_{2i}^{\mathrm{null}} + \omega(\tilde{\mathbf{z}}_{2i}^{\mathrm{cond}} - \tilde{\mathbf{z}}_{2i}^{\mathrm{null}})$
7:   $\mathbf{z}_i \leftarrow \mathrm{Concat}(\tilde{\mathbf{z}}_{2i-1}, \tilde{\mathbf{z}}_{2i})$
8:   **if** $\mathtt{mode} = \mathtt{Diffusion}$ **then**
9:    $\mathbf{x}_i \leftarrow \mathrm{DiffusionSampling}(D_\theta, \mathbf{z}_i)$      $\triangleright$ Multi-step sampling
10:   **else**
11:    $\mathbf{x}_i \leftarrow \mathrm{CMSampling}(D_\theta, \mathbf{z}_i)$     $\triangleright$ Few-step sampling (NFE=1~4)
12:   **end if**
13:   **if** $\mathtt{streaming\_decode}$ and $\mathrm{Dec}_{\mathrm{VAE}}$ is causal **then**
14:    $\hat{\mathbf{y}}_{1:i} \leftarrow \mathrm{Dec}_{\mathrm{VAE}}(\{\mathbf{x}_j\}_{j=1}^i)$       $\triangleright$ Incremental decoding
15:   **end if**
16: **end for**
17: **if not** $\mathtt{streaming\_decode}$ **then**
18:   $\hat{\mathbf{y}}_{1:n} \leftarrow \mathrm{Dec}_{\mathrm{VAE}}(\{\mathbf{x}_i\}_{i=1}^n)$      $\triangleright$ Decode all tokens at once
19: **end if**
20: **return** Waveform $\hat{\mathbf{y}}_{1:n}$

---

aggregator, we use a single non-causal transformer layer with learnable positional embeddings. We conduct ablation studies on these vision conditions in Appendix C.3.

**Audio token modeling** We follow the stereo VAE from SA Series (Evans et al., 2025; 2024) as a model backbone. We change the temporal downsampling rate from 2048 to 1600 and train with 48 kHz stereo waveform by using OGameData250K and WavCaps (Mei et al., 2024a) dataset from scratch. We follow the original training configuration (Evans et al., 2025). The network parameter size is 157 M. In the latent space, we get 64 dimensions of 30-Hz continuous audio latents.

**Multimodal transformer with diffusion head** For the multimodal AR transformer, we use 18 layers, 1024 hidden dimensions, and 16 heads. The model backbone follows LLaMA-style design (Touvron et al., 2023a;b), applying pre-normalization using RMSNorm (Zhang & Sennrich, 2019), SwiGLU activation (Shazeer, 2020), and rotary positional embeddings (RoPE) (Su et al., 2024). For the diffusion head, we set 8 blocks and a width of 1280 channels. The architecture of the diffusion head largely follows the original architecture in MAR. The total parameter size is 320 M (250 M for the transformer and 70 M for the diffusion head). We ablate various numbers of parameters of the diffusion head in Appendix C.1.

### B.2 TRAINING DETAILS

**Stage-1: Diffusion pretraining** After obtaining $\mathbf{x}^0$ through the audio VAE, $\mathbf{x}^0$ is standardized globally to zero mean and standard deviation $\sigma_{\mathrm{data}} = 0.5$ by following EDM2 (Karras et al., 2024). We utilize the EDM's skip connection $c_{\mathrm{skip}}(t)$, output scale $c_{\mathrm{out}}(t)$, input scale $c_{\mathrm{in}}(t)$, and $c_{\mathrm{noise}}(t)$ for modeling $D_\theta$ as:

$$D_\theta(\mathbf{x}_i^t, t, \mathbf{z}_i) = c_{\mathrm{skip}}(t)\,\mathbf{x}_i^t + c_{\mathrm{out}}(t)\,\mathrm{NN}_\theta\big(c_{\mathrm{in}}(t)\,\mathbf{x}_i^t, c_{\mathrm{noise}}(t), \mathbf{z}_i\big), \tag{9a}$$

$$c_{\mathrm{skip}}(t) = \sigma_{\mathrm{data}}^2/(t^2 + \sigma_{\mathrm{data}}^2), \tag{9b}$$

$$c_{\mathrm{out}}(t) = (t\,\sigma_{\mathrm{data}})/\sqrt{t^2 + \sigma_{\mathrm{data}}^2}, \tag{9c}$$

$$c_{\mathrm{in}}(t) = 1/\sqrt{t^2 + \sigma_{\mathrm{data}}^2}, \tag{9d}$$

$$c_{\mathrm{noise}}(t) = \tfrac{1}{4}\ln t, \tag{9e}$$

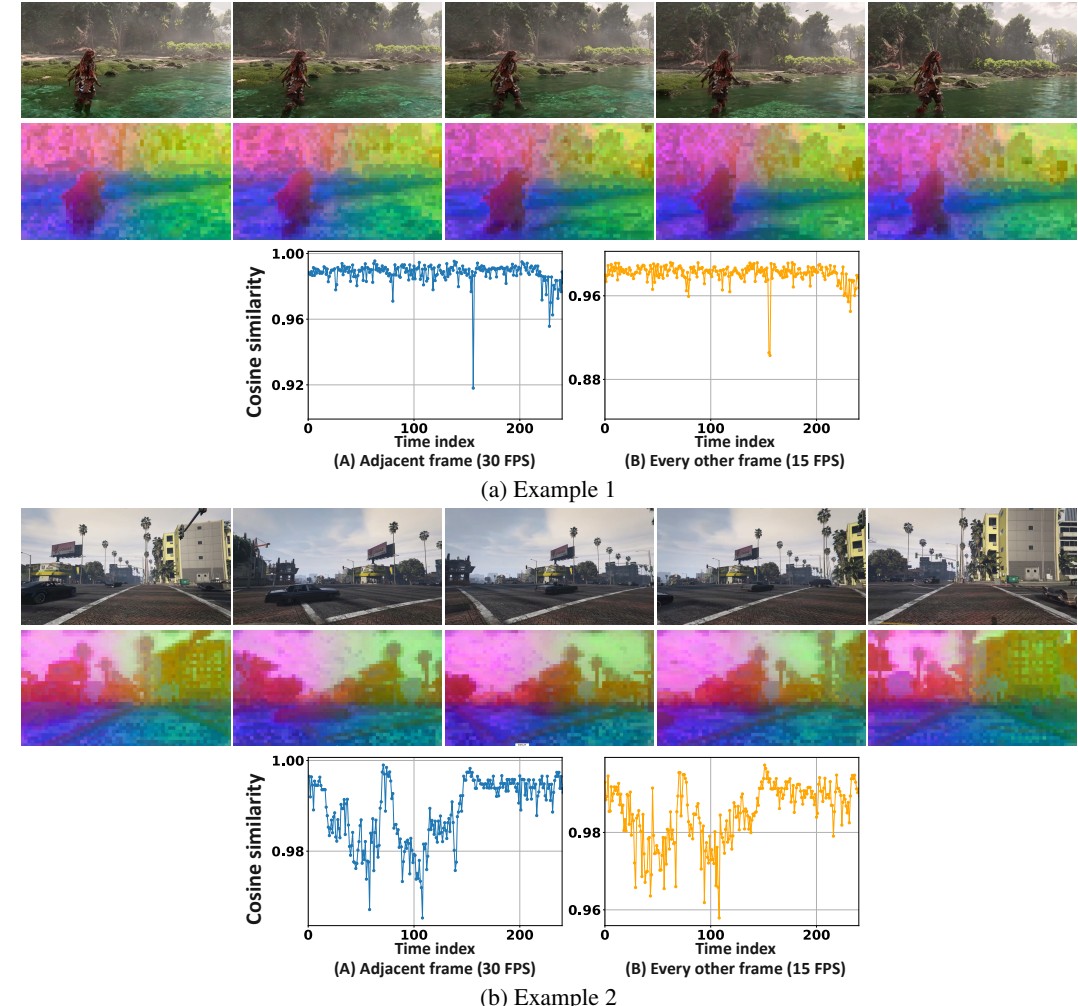

Figure 5: Visualization of DINOv2 grid features and cosine similarity of their [`CLS`]-tokens between (A) adjacent frames and (B) every other frame. Visualizations are done by applying PCA to grid features and mapping them to RGB.

where $\mathrm{NN}_{\boldsymbol{\theta}}$ is the actual neural network to be trained. We set $P_{\mathrm{mean}} = -0.4$, $P_{\mathrm{std}} = 1.0$ for a training noise sampling $\ln t \sim \mathcal{N}(P_{\mathrm{mean}}, P_{\mathrm{std}})$ and network dropout rate of $10\%$ (Srivastava et al., 2014). The models are trained using the AdamW optimizer (Loshchilov & Hutter, 2019) for 300K training iterations. Training is done on 8×NVIDIA H100 GPUs. Training duration is around 36 hrs. Training is done with bf16. The weight decay and momenta for AdamW are $0.02$ and $(0.9, 0.95)$. We set a learning rate of $1e - 4$ with the constant learning rate scheduler, EMA rate of $0.9999$, and gradient clipping as $1.0$. By default, we use a batch size of $128$. To enable CFG sampling on the transformer, during training, we randomly replace all the video tokens $\mathbf{v}_i (i = 1, \ldots, n)$ with the learnable null embedding with the probability of $10\%$. As our denoising network is not so large, we can sample $t$ multiple times for any given $\mathbf{z}_i$. We sample $t$ by 4 times during training for each sample by following (Li et al., 2024). This operation helps improve the utilization of the loss function.

**Stage-2: ECT**  As same as Stage 1 network configuration, we utilize the EDM's skip connection $c_{\mathrm{skip}}(t)$, output scale $c_{\mathrm{out}}(t)$, input scale $c_{\mathrm{in}}(t)$, and $c_{\mathrm{noise}}(t)$ for modeling the head. For a metric function $d$, we use a Pseudo-Huber loss $d(\mathbf{a}, \mathbf{b}) = \sqrt{\|\mathbf{a} - \mathbf{b}\|_2^2 + \nu^2} - \nu$, where $\nu = 0.06$ (Song & Dhariwal, 2024; Geng et al., 2025b). By following the original work, we set $P_{\mathrm{mean}} = -0.8$, $P_{\mathrm{std}} = 1.6$ for a training noise sampling $\ln t \sim \mathcal{N}(P_{\mathrm{mean}}, P_{\mathrm{std}})$, and $w(t) = 1/t^2 + 1/\sigma_{\mathrm{data}}^2$. For the mapping function $m(r \mid t, \mathrm{Iters})$ (in Eq 8), we explore two different setups suggested by the literature (Geng et al., 2025b), which we describe in Section 4.5. For CF (our default), we set $q = 2$, $s = \mathrm{total\_iter}//8$, $b = 1$, $k = 8$ in Eq (8). For IN, we set $q = 4$, $s = \mathrm{total\_iter}//4$, $b = 1$,

$k = 8$. The models are trained using the AdamW optimizer (Loshchilov & Hutter, 2019) for 200K training iterations. Training is done on 8×NVIDIA H100 GPUs. Training duration is around 20 hours. Training is done with bf16. The weight decay and momenta for AdamW are $0.02$ and $(0.9, 0.95)$. We set a learning rate of $1e-4$ with the constant learning rate scheduler, EMA rate of $0.9999$ for $\mathrm{sg}(\boldsymbol{\theta}, \boldsymbol{\phi})$, batch size of 128, and gradient clipping as 1.0. To enable CFG sampling, we randomly replace the learnable null embedding with all the video tokens $\mathbf{v}_i (i = 1, \ldots, n)$ with the probability of 10%. We sample $t$ by 4 times during training for each sample by following the Stage 1 training.

### B.3 SAMPLING DETAILS

**AR transformer**    Our sampling procedure follows that of a standard decoder-only AR transformer with a token-wise KV cache (Shazeer, 2019). As described in Section 3.4, to get $\tilde{\mathbf{z}}_{2i}$ (a corresponding input token is a vision condition $\mathbf{v}_i$), we apply CFG as $\tilde{\mathbf{z}}_{2i} = \tilde{\mathbf{z}}_{2i}^{\mathrm{null}} + \omega(\tilde{\mathbf{z}}_{2i}^{\mathrm{cond}} - \tilde{\mathbf{z}}_{2i}^{\mathrm{null}})$, where $\omega$ is a guidance scale, $\tilde{\mathbf{z}}_{2i}^{\mathrm{cond}}$ is the transformer output given the visual condition $\mathbf{v}_i$, and $\tilde{\mathbf{z}}_{2i}^{\mathrm{null}}$ is the output given a learnable null embedding, respectively. To get $\tilde{\mathbf{z}}_{2i}^{\mathrm{null}}$, all the $\mathbf{v}_{\leq i}$ are replaced with the learnable null embedding. Sampling is done with bf16.

Our approach contrasts with the CFG strategy in the original MAR (Li et al., 2024), where CFG is applied within the diffusion head. In the original MAR, the transformer generates both conditional $\mathbf{z}_i^{\mathrm{cond}}$ and unconditional $\mathbf{z}_i^{\mathrm{uncond}}$ vectors, which are then used for the standard CFG diffusion sampling. In contrast, we do not apply CFG within the diffusion head. We compare the performance of both CFG strategies in Appendix C.1. Our design choice has a practical benefit, except for the performance: it reduces the computational overhead during sampling on the diffusion head, which might be critical when the head becomes larger or using the head without sampling acceleration.

**Diffusion head**    For the diffusion sampling on our Stage-1 model, we use the deterministic Heun solver with the same time-step discretizations as EDM (Karras et al., 2022; 2024). Namely, if we generate samples with $N$-steps, $t_j = (t_{\max}^{1/\rho} + \frac{j}{N-1}(t_{\min}^{1/\rho} - t_{\max}^{1/\rho}))^\rho$, where $\rho = 7$. On our Stage-2 model, we set intermediate time step $t = 2.5$ and $t = [5.0, 1.1, 0.08]$ on NFE=2 and NFE=4, respectively. We use the EMA weights during sampling and set $\{t_{\max}, t_{\min}\} = \{80, 0.00\}$ on the both Stage-1 and Stage- 2 (Karras et al., 2022; 2024; Song & Dhariwal, 2024; Geng et al., 2025b). Sampling is done with bf16. We provide the sampling procedure in Algorithm 3.

### B.4 CONTEXT WINDOW EXTENSION ON ROPE

In Section 4.4, we test our model on longer sequence generation beyond the training context window up to twice the trainig window on SoundReactor-ECT with NFE=4. In the experiments, we examine two zero-shot context window extension methods based on RoPE, Position Interpolation (PI) (Chen et al., 2023) and NTK-aware interpolation (NTK) (Tancik et al., 2020), and Sliding Window Attention (SWA) (Beltagy et al., 2020; Jiang et al., 2023). Here, we describe those methods.

RoPE is applied to both the query ($\mathbf{q}$) and key ($\mathbf{k}$) vectors within a transformer layer. An input token $\mathbf{z}_i$ is first projected into a query vector $\mathbf{q}_i = W_q \mathbf{z}_i$ and a key vector $\mathbf{k}_i = W_k \mathbf{z}_i$, where $\mathbf{q}_i, \mathbf{k}_i \in \mathbb{R}^{c_z}$. To encode the positional information, the $\mathbf{q}_i$ is treated as a sequence of two-dimensional vectors $(q_{i,2l-1}, q_{i,2l})$, where $l$ is the index ranging from 1 to $c_{\mathbf{z}}/2$. A rotation is then applied to each pair as follows:

$$\begin{pmatrix} q'_{i,2l-1} \\ q'_{i,2l} \end{pmatrix} = \begin{pmatrix} \cos i\varphi_l & -\sin i\varphi_l \\ \sin i\varphi_l & \cos i\varphi_l \end{pmatrix} \begin{pmatrix} q_{i,2l-1} \\ q_{i,2l} \end{pmatrix}, \tag{10}$$

where the frequency $\varphi_l = \xi^{-2l/c_{\mathbf{z}}}$ depends on the index $l$ and a base $\xi$. The same process is applied to the key vector $\mathbf{k}_i$.

**Position interpolation (PI)**    PI adapts RoPE to longer sequences by down-scaling the position indices via $i' = i \cdot (n/n')$, where $n$ and $n'$ are a training sequence length and a target sequence length during inference, respectively. This operation uniformly compresses the entire frequency spectrum, which can degrade the model's understanding of high-frequency details essential for precise temporal alignment.

**NTK-aware interpolation (NTK)**   Instead of scaling the position indices, NTK scales the base $\xi$ of the frequency calculation to a new value $\xi' = \xi \cdot (n'/n)^{c_\mathbf{z}/(c_\mathbf{z}-2)}$. The approach is motivated by Neural Tangent Kernel theory (Tancik et al., 2020), which posits that deep neural networks have trouble learning high-frequency information if the input dimension is low without the corresponding embeddings having high-frequency components. NTK's base rescaling allows the model to retain its modeling capabilities on high-frequency information, which we qualitatively observe in Figure 3.

**Sliding window attention (SWA)**   Besides extending the context window, SWA is one of the standard and efficient attention mechanisms that restricts each token to only attend to a fixed-size local window of preceding tokens. For generation beyond the training context window, it simply continues to apply this local attention window. While this approach is computationally efficient, its ability to model long-range dependencies is inherently limited by the fixed window size.

**Discussion**   In addition to quantitative evaluation in Section 4.4, the spectrogram visualization in Figure 3 shows that PI slows periodic sounds (e.g., footsteps) and harms temporal synchronization, while NTK and SWA preserve timing. We conjecture that this stems from how RoPE frequencies are scaled. Our interleaved, frame-aligned audio–visual tokens rely on high-frequency positional components for the precise timing of action sounds. NTK rescales the RoPE base and preserves high-frequency structure via $\xi' = \xi \cdot (n'/n)^{c_\mathbf{z}/(c_\mathbf{z}-2)}$. In contrast, PI stretches positions in the time domain $i' = i \cdot (n/n')$, which lowers the angular frequency. This experiment is complementary to prior findings from the other domains, such as language domain, by demonstrating the superiority of NTK over PI in a multimodal audio-visual setting. We validate its effectiveness through not only the quantitative performance difference but also the qualitative analysis of the spectrograms.

### B.5   PER-FRAME LATENCY MEASUREMENT

We measure two metrics: wave-level and token-level latency. Wave-level latency is the elapsed time from feeding the previous audio token $\mathbf{x}_{n-1}$ into the model until the output $\mathbf{x}_n$ is incrementally decoded into a waveform, including the encoding of the raw video frame $V_n$. Token-level latency is identical to this but excludes the incremental waveform decoding. Following Algorithm 3, we insert `torch.cuda.synchronize()` calls between lines 2–3 and 14–15 to measure wave-level latency, and between lines 2–3 and 11–12 to measure token-level latency. This duration is recorded as the per-frame latency. For each video, we average the per-frame latencies over all frames except the initial one. We perform this benchmark on a single H100 GPU with a batch size of one, using a single CUDA stream for the entire pipeline. We use 53 clips (30FPS, 480p, 16 second) with the default SoundReactor-ECT at $\omega = 3$. We use NTK to generate 16 seconds with KV-cache[10]. We apply FlashAttention-2 (Dao, 2024) for the main model. The `torch.compile` is enabled on the main model while CUDA Graph is used on the VAE decoder. To measure the wave-level latency, we use a causal stereo full-band VAE, which is created by fine-tuning our default VAE to make the decoder causal while freezing the encoder parameters.

We provide the mean and standard deviation over the 50 clips following a 3-clip warm-up period[11]. As demonstrated in Table 6, while our model achieves remarkably low per-frame token-level latency, the overhead from incremental waveform decoding prevents the end-to-end wave-level latency from meeting the 33.3 ms threshold (per-frame real-time for 30FPS video). In terms of sample quality, we observe that the causal VAE decoder lags behind its non-causal counterpart in reconstruction quality (causal: $\text{rMMD}_{\text{OpenL3}} = 62.7$, $\text{rMMD}_{\text{CLAP}} = 0.538$; non-causal: $\text{rMMD}_{\text{OpenL3}} = 60.4$, $\text{rMMD}_{\text{CLAP}} = 0.301$). This performance gap propagates to final generation quality: e.g., our model on ECT (NFE=4) with the causal decoder yields $\text{MMD}_{\text{OpenL3}} = 65.5$ and $\text{MMD}_{\text{CLAP}} = 0.682$, versus $64.6$ and $0.451$ with the non-causal one (our default setup). Accordingly, addressing the causal VAE's decoding overhead and fidelity gap remains an important direction for our future work.

### B.6   METRICS DETAILS

**Audio quality**   To evaluate audio quality, we use Frechet Audio Distance (FAD) (Gui et al., 2024), Maximum Mean Discrepancy (MMD) (Chung et al., 2025; Jayasumana et al., 2024), and

---

[10]NTK, PI, and no RoPE extension do not affect latency.

[11]This is for excluding system warm-up effects, such as GPU kernel initialization.

Kullback-Leibler divergence based on PaSST (Koutini et al., 2022) (KL$_{\text{PaSST}}$). For FAD and MMD, following prior work Evans et al. (2024; 2025); Gui et al. (2024), we use OpenL3 (Cramer et al., 2019) and LAION-CLAP (Wu* et al., 2023). For OpenL3, we use (Env, Mel256, 512) in `https://github.com/torchopenl3/torchopenl3`. For CLAP, we use the `630k-audioset-fusion-best.pt` from `https://github.com/LAION-AI/CLAP` as audio feature extractors. Since our model is trained on stereo signals, we compute the metrics on four channel configurations: the left channel (Left), the right channel (Right), their average (Center), and their difference (Diff) (Steinmetz et al., 2021). We compute KL$_{\text{PaSST}}$ only on the Center. For the samples from the models trained on monaural signal, they are duplicated for the left and right channels. Consequently, the Diff is omitted as the difference becomes zero. Furthermore, as a complementary metric to evaluate stereo quality, we also use Frechet Stereo Audio Distance (FSAD) (Sun et al., 2025), which is based on StereoCRW (Chen et al., 2022b) features.

**Audio-visual alignment** To evaluate audio-visual semantic and temporal alignment, we utilize ImageBind score (IB-Score) (Girdhar et al., 2023) and DeSync (Viertola et al., 2025) by following common practices (Cheng et al., 2025; Viertola et al., 2025). To compute DeSync, we take two crops (first 4.8 sec. and last 4.8 sec.) from the video and compute the average over the results (Cheng et al., 2025). We compute these two metrics only on the Center.

# C  ADDITIONAL EXPERIMENTS

## C.1  ABLATION STUDY ON GENERATOR TRAINING

**Diffusion head size** In Table 7, we analyze the capacity of the diffusion head while keeping the total model parameters fixed at 320 M parameters. We use 35 (60%CEV) for this experiment. A larger head consistently yields better generation quality. Notably, while the original MAR generates valid images even with very small heads (e.g., 2M and 6M param-

Table 7: Ablation study on various sizes of diffusion head on SoundReactor-Diffusion under a fixed entire network capacity. Bold and underlined scores indicate the best and second-best results, respectively.

| Head size | MMD↓ | | | | KL$_{\text{PaSST}}$ ↓ | IB-Score ↑ | DeSync ↓ |
|---|---|---|---|---|---|---|---|
| | OpenL3 | | CLAP | | | | |
| | Center | Diff | Center | Diff | | | |
| 10M | 76.4 | 45.0 | 0.943 | 1.44 | 1.66 | 0.277 | 1.06 |
| 30M | **68.3** | 37.0 | 0.684 | 1.31 | **1.56** | **0.280** | **1.03** |
| 50M | 68.4 | 35.5 | 0.628 | **1.25** | **1.56** | **0.280** | 1.04 |
| 70M (default) | 69.3 | **35.1** | **0.597** | 1.29 | **1.56** | **0.280** | **1.03** |

eters) (Li et al., 2024), our 10M head variant failed to produce high-quality samples. We hypothesize this is due to the challenge of handling high-dimensional audio tokens ($c_{\mathbf{x}} = 64$)[12]. Indeed, our preliminary trials on an even higher dimension of $c_{\mathbf{x}} = 128$ yielded no valid audio even with a 50 M head. Consequently, the necessity of allocating substantial capacity to the diffusion head directly slows down decoding speed per token, providing a strong motivation for the step-wise acceleration that we introduce in this work.

**ECT strategy regarding fine-tuning component** We conduct an ablation study on two ECT fine-tuning strategies: fine-tuning only the diffusion head and fine-tuning the entire network (our default setup). As shown in Table 8, fine-tuning the entire network yields superior overall performance. However, we newly find that, interestingly, fine-tuning only the diffusion head is still

Table 8: ECT fine-tuning with the entire network (Our default) and only with the diffusion head (Only head). We use `IN-0.2`. MMD is computed on Center.

| Model | MMD↓ | | KL$_{\text{PaSST}}$ ↓ | IB-Score ↑ | DeSync ↓ |
|---|---|---|---|---|---|
| | OpenL3 | CLAP | | | |
| Only head | | | | | |
| `IN-0.2` (NFE=2) | 68.7 | 0.623 | 1.59 | 0.272 | 1.05 |
| `IN-0.2` (NFE=4) | 62.1 | 0.585 | 1.61 | 0.272 | **1.03** |
| **Our default** | | | | | |
| `IN-0.2` (NFE=2) | 71.65 | 0.445 | 1.57 | **0.284** | 1.03 |
| `IN-0.2` (NFE=4) | 59.79 | 0.411 | 1.56 | 0.280 | 1.03 |

sufficient to generate good audio samples. This insight suggests a potential path toward more efficient step-wise acceleration on the diffusion heads.

## C.2  ABLATION STUDY ON SAMPLING STRATEGY

Here, we analyze the model's behavior with various sampling strategies. Figure 6 shows the objective metrics for SoundReactor-Diffusion with various numbers of sampling steps, with the fixed guidance

---

[12]A similar discussion has been reported in `https://github.com/LTH14/mar/issues/55`.

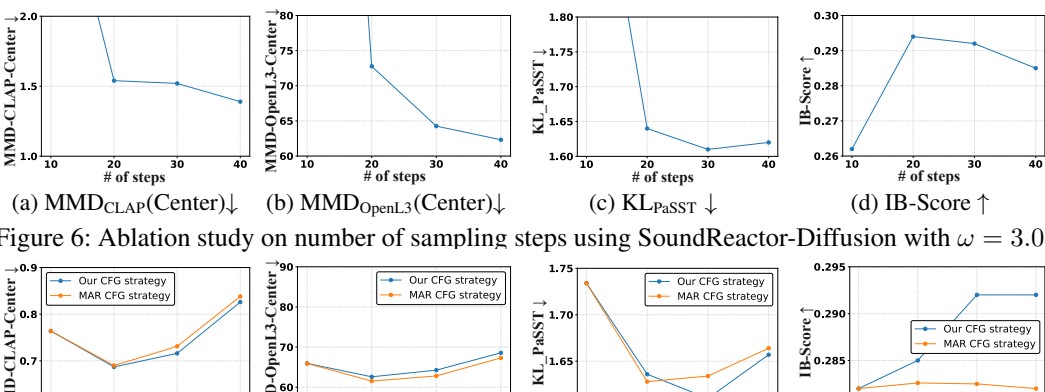

Figure 6: Ablation study on number of sampling steps using SoundReactor-Diffusion with $\omega = 3.0$.

(a) MMD$_{\text{CLAP}}$(Center)↓   (b) MMD$_{\text{OpenL3}}$(Center)↓   (c) KL$_{\text{PaSST}}$ ↓   (d) IB-Score ↑

Figure 7: Ablation study on CFG scale $\omega$ using SoundReactor-Diffusion with 30 steps. Differences between our strategy and MAR's are described in Appendix B.3.

scale $\omega = 3.0$. We observe that 10 steps (NFE=19) fail to generate valid samples, 20 steps (NFE=39) begin to yield reasonable samples, and 30 (NFE=59) or more steps are required for high-quality samples. These results motivate us to conduct diffusion head acceleration to achieve low per-frame latency.

In Figure 7, we fix the number of steps to 30 and examine the impact of the CFG scale $\omega$. The results indicate an optimal performance range between $\omega = 2.0$ and $\omega = 3.0$. We also include results from applying the original MAR's CFG strategy to our models. Concretely, $\mathbf{z}_i^{\text{cond}} = \text{Concat}(\tilde{\mathbf{z}}_{2i-1}, \tilde{\mathbf{z}}_{2i}^{\text{cond}})$ and $\mathbf{z}_i^{\text{uncond}} = \text{Concat}(\tilde{\mathbf{z}}_{2i-1}, \tilde{\mathbf{z}}_{2i}^{\text{null}})$ are used to conduct CFG sampling on the diffusion head (See the difference from our strategy explained in Algorithm 3 and Appendix B.3). While both strategies perform comparably, our approach is more computationally efficient, as noted in Appendix B.3, since our strategy does not use CFG on either diffusion or CM sampling.

Figure 8 presents the objective metrics for SoundReactor-ECT across various NFEs and $\omega$ values. Similar to the results for SoundReactor-Diffusion, we observe a consistent performance sweet spot around $\omega = 2.0$ and $\omega = 3.0$ for all evaluated NFEs on the head.

## C.3   ABLATION STUDY ON VISION CONDITIONING

**Influence of dimensionality reduction via PCA**   To investigate the influence of dimensionality reduction on the pretrained grid features by PCA (See Appendix B.1), we train the models with feature dimensions of 35, 59, and 99 dimensions, corresponding to CEV levels of 60%, 70%, and 80%, respectively. All models in this specific experiment are trained with half the default batch size. As shown in Table 9, interestingly, even the compressed representation achieves better results on several metrics

Table 9: Varying dimensionality of PCA-reduced features and the impact of using our adjacent frame subtraction. Tested on SoundReactor-Diffusion. Bold and underlined scores indicate the best and second-best results, respectively.

| Channel-dim | MMD↓ | | KL$_{\text{PaSST}}$ ↓ | IB-Score ↑ | DeSync ↓ |
|---|---|---|---|---|---|
| | OpenL3 | CLAP | | | |
| 35 (60% CEV) | **62.8** | 0.669 | 1.79 | 0.281 | 1.07 |
| w/o. subtraction | 71.2 | 0.697 | 1.60 | 0.290 | **1.04** |
| 59 (default) (70% CEV) | 63.9 | 0.666 | 1.60 | 0.285 | 1.06 |
| w/o. subtraction | 74.7 | 0.831 | 1.75 | 0.292 | 1.07 |
| 99 (80% CEV) | 67.4 | **0.655** | **1.59** | 0.284 | **1.04** |
| w/o. subtraction | 72.1 | 0.715 | 1.61 | **0.293** | 1.05 |

compared to its higher-dimensional counterparts. Although the dimensionality necessary for modeling is expected to vary across data distributions, our empirical results suggest that vision conditioning might be feasible with compact representations derived from a selected subset of salient features.

**Effectiveness of adjacent frame subtraction**   We ablate our vision conditioning mechanism that utilizes temporal differences between adjacent grid features (described in Section 3.2). As demonstrated in Table 9, removing this component consistently degraded performance. This indicates that incorporating temporal information via frame differencing is a crucial component of our vision conditioning.

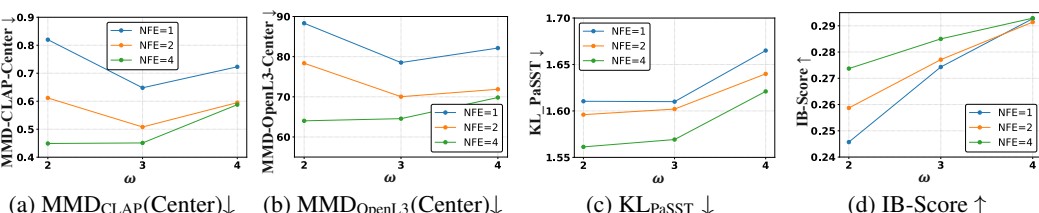

(a) MMD$_{\text{CLAP}}$(Center)↓    (b) MMD$_{\text{OpenL3}}$(Center)↓    (c) KL$_{\text{PaSST}}$ ↓    (d) IB-Score ↑

Figure 8: Ablation study on CFG scale $\omega$ using SoundReactor-ECT with various NFEs on head.

## C.4 SUPPLEMENTARY EXPERIMENTS ON VGGSOUND

### C.4.1 EXPERIMENTAL SETUP

As a supplementary evaluation, we benchmark our framework against various offline V2A approaches on the VGGSound dataset (Chen et al., 2020b). VGGSound consists of 200K+ 10-second (around 500 hours) real-world uncurated video clips from YouTube spanning 309 categories. We follow the data split and preprocessing pipeline in MMAudio (Cheng et al., 2025), where the training set contains approximately 180K 8-second videos (around 400 hours). We use only audio-visual pairs, without their corresponding class labels. All data is processed at 30 FPS with $640 \times 360$ resolution and 48kHz stereo audio. For evaluation, consistent with previous work (Cheng et al., 2025; Zhong et al., 2025), we use 8-second clips from the VGGSound test set (15K videos).

For objective metrics, we use Frechet Audio Distance (FAD$_{\text{PaSST}}$) (Gui et al., 2024) and Kullback–Leibler divergence (KL$_{\text{PaSST}}$) based on PaSST (Koutini et al., 2022), a 32 kHz state-of-the-art audio classifier. For audio-visual alignment, we use ImageBind score (IB-Score) (Girdhar et al., 2023) and DeSync (Viertola et al., 2025), which is computed on the features extracted by Synchformer. We only compute these metrics on Center, derived by averaging the left and right channels.

For the model training, we largely follow the default setup from our OGameData experiments (detailed in Appendix B), with the following exceptions: we use a three-layer non-causal transformer for the aggregator (instead of a single layer), a 768-dimensional learnable token (instead of 512-dimensional), and reduce the DINOv2 (dinov2-vits14-reg) grid feature dimension to 146 via PCA, retaining the CEV of 80%. For inference, we set $\omega = 4.0$ across all experiments and use a 30-step deterministic Heun Solver for SoundReactor-Diffusion. For SoundReactor-ECT, we set intermediate time step $t = 2.5$ and $t = [5.0, 1.1, 0.08]$ on NFE=2 and NFE=4, respectively. We use the causal stereo full-band VAE, which is created by fine-tuning our default VAE to make both the encoder and decoder causal, for this experiment.

### C.4.2 RESULTS

The results are presented in Table 10, with details of the baseline methods in Appendix A.1. Compared to the offline AR model, V-AURA, our framework shows a similar trend on FAD$_{\text{PaSST}}$ and DeSync as in the OGameData experiments, though it lags behind on IB-Score and KL$_{\text{PaSST}}$. Against offline Non-AR models, our framework outperforms several methods such as Seeing & Hearing, ReWaS, and FoleyCrafter, while a significant performance gap remains with MMAudio, which is the state-of-the-art V2A model[13]. Overall, these results demonstrate that our framework is generalizable to real-world video datasets, despite some performance limitations. Demo samples of the VGGSound are available at https://anonymous-sr-submission.github.io/vggsound.html.

Potential avenues for bridging this performance gap include using larger image encoders such as CLIP-L (300M) and CLIP-H (840M) that used in the baselines (or the larger variants of DINOv2) or employing video encoders pre-trained on large-scale audio-visual datasets, which can extract video features under a causal constraint. However, designing effective architectures that incorporate these components while preserving the end-to-end causality and low frame-level latency, which are crucial for the online V2A task, is a non-trivial challenge. Therefore, we leave this as future work.

---

[13]We also observe that the DeSync appears to favor models that use Synchformer features, such as SpecMask-Foley, V-AURA, and MMAudio, which outperform the VAE reconstruction, suggesting a potential methodological bias.

Table 10: Additional experiments on VGGSound test set with various offline V2A models. Only our models are applicable to frame-level online V2A task, which is our focus on this work. Following common practice (Cheng et al., 2025), parameter counts only on generative modeling part.

| Model | Params | Channels/ Sample rate | Text condition | FAD$_{PaSST}$ ↓ | KL$_{PaSST}$ ↓ | IB-Score ↑ | DeSync ↓ |
|---|---|---|---|---|---|---|---|
| Our VAE-Reconstruction | – | 2/48kHz | – | 57.0 | 0.361 | 27.0 | 0.70 |
| **Offline AR models** | | | | | | | |
| V-AURA (Viertola et al., 2025) | 695M | 1/44.1kHz | ✗ | 219 | 2.07 | 0.28 | 0.65 |
| **Offline Non-AR models** | | | | | | | |
| Frieren (Wang et al., 2024b) | 159M | 1/16kHz | ✗ | 106 | 2.86 | 0.23 | 0.85 |
| V2A–Mapper (Wang et al., 2024a) | 229M | 1/16kHz | ✗ | 84.6 | 2.56 | 0.23 | 1.23 |
| SpecMaskFoley (Zhong et al., 2025) | 300M | 1/22.05kHz | ✓ | 109 | 1.76 | 0.26 | 0.65 |
| Seeing & Hearing (Xing et al., 2024) | 415M | 1/16kHz | ✓ | 219 | 2.30 | **0.34** | 1.20 |
| ReWaS (Jeong et al., 2024) | 620M | 1/16kHz | ✓ | 141 | 2.82 | 0.15 | 1.06 |
| MMAudio-L (Cheng et al., 2025) | 1.03B | 1/44.1kHz | ✓ | **60.6** | **1.40** | 0.33 | **0.44** |
| FoleyCrafter (Zhang et al., 2024) | 1.22B | 1/16kHz | ✓ | 140 | 2.23 | 0.26 | 1.23 |
| **Frame-level online models (Ours)** | | | | | | | |
| SoundReactor-Diffusion | 336M | 2/48kHz | ✗ | 175 | 2.37 | 0.21 | 1.04 |
| SoundReactor-ECT (NFE=1) | 336M | 2/48kHz | ✗ | 194 | 2.37 | 0.21 | 1.03 |
| SoundReactor-ECT (NFE=2) | 336M | 2/48kHz | ✗ | 130 | 2.32 | 0.24 | 1.02 |
| SoundReactor-ECT (NFE=4) | 336M | 2/48kHz | ✗ | 113 | 2.37 | 0.24 | 1.03 |

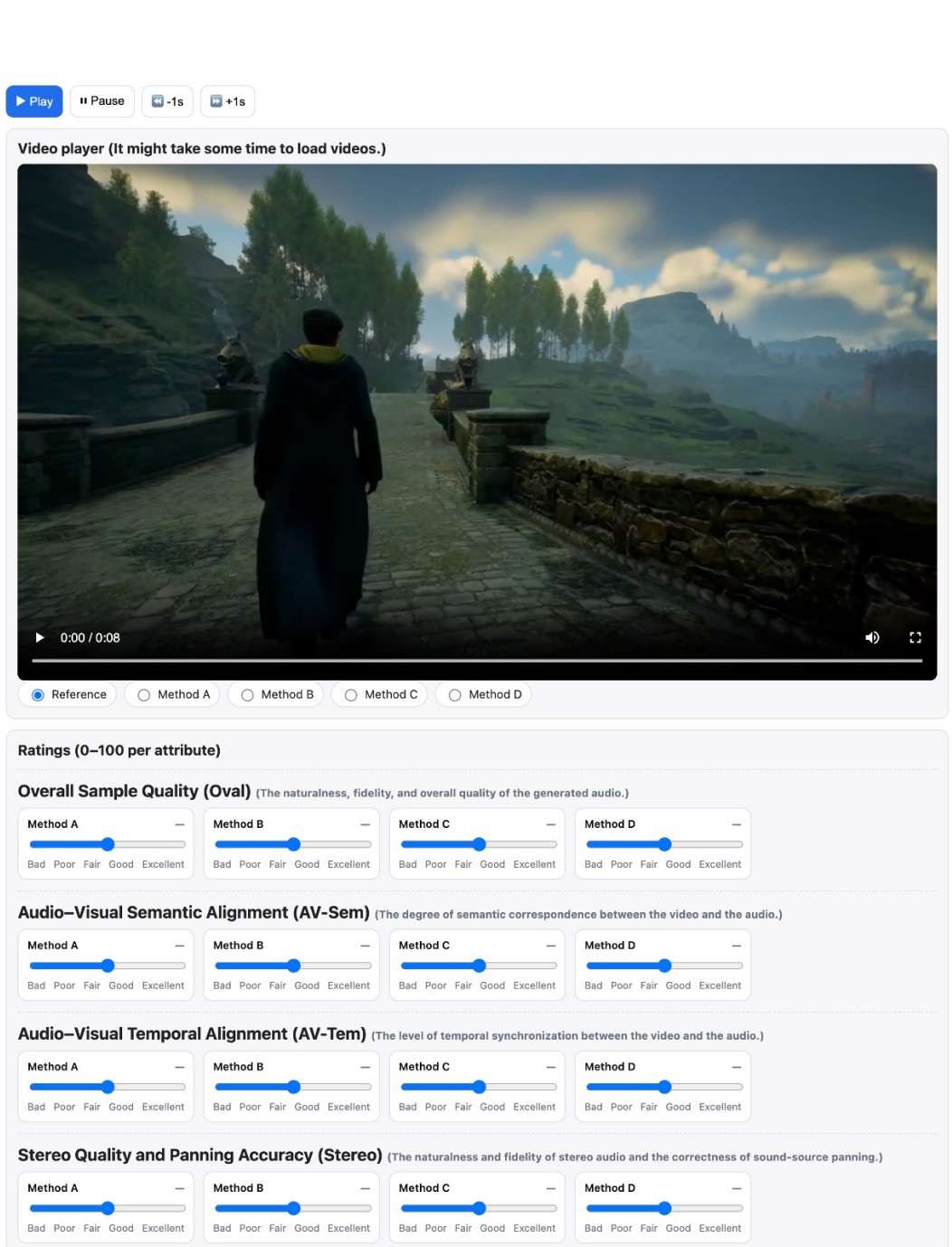

Figure 9: Screenshot of human evaluation

