# OpenReview forum: "SoundReactor: Frame-level Online Video-to-Audio Generation"
_ICLR.cc/2026/Conference — ICLR 2026 Conference Withdrawn Submission_

### Official Review · Reviewer_ygRm · 2025-10-30

**Soundness:** 4
**Presentation:** 4
**Contribution:** 3
**Rating:** 6
**Confidence:** 3

**Summary:**

The paper proposes SoundReactor, a novel framework for frame-level online video-to-audio (V2A) generation. To enforce causality, the model employs a causal decoder-only transformer with a diffusion head on top. To improve generation speed, the authors apply Easy Consistency Tuning (ECT). For conditioning, visual features are extracted from DINOv2 grid representations, while audio tokens are obtained from a VAE trained from scratch. The model demonstrates strong performance and low latency compared to the baseline V-AURA on gameplay videos.

**Strengths:**

1.	This is the first work to explicitly address a novel online, frame-level V2A generation, and the design choices are technically sound.
2.	The model achieves both high efficiency and strong performance in OGameData.
3.	The experiments are extensive, with comprehensive ablations supporting the design choice.

**Weaknesses:**

1.	Real-time generation is most impactful in long or interactive settings. However, the experiments are limited to 16-second clips, which may not convincingly demonstrate online efficiency, since prior works (e.g., SpecVQGAN) already handle sequences of ~10 s or more.
2.	It would be helpful to include an ablation using a non-causal or offline transformer to quantify how much the causal constraint actually contributes whether positively or negatively to the overall performance.
3.	Given the frame-level generation setup, does the model also produce a long silence when no visual events occur? This aspect is not clearly discussed.

**Questions:**

1.	Could this framework be extended to multi-condition generation (e.g., adding text prompts to control style or sound context)?
2. If a short irrelevant segment (e.g., a 1-second advertisement) appears before the actual video (e.g., 8 seconds of a bird singing), will the subsequent audio generation be influenced by the earlier context? How quickly can the model adapt to such context shifts?

---

### Official Review · Reviewer_29S2 · 2025-10-31

**Soundness:** 3
**Presentation:** 3
**Contribution:** 3
**Rating:** 2
**Confidence:** 3

**Summary:**

The paper, SoundReactor, introduces the first frame-level online Video-to-Audio (V2A) generation model that can generate synchronized audio in real time from video frames. Unlike prior offline V2A models, which require access to the entire video frames before producing audio, SoundReactor operates causally, generating audio sequentially, one frame at a time, without future frames. The framework consists of three core components: (1) Video token modeling; (2) Audio token modeling; (3) Multimodal transformer with diffusion head Training is performed in two stages: first, diffusion pretraining under the EDM2 framework to model robust denoising and temporal coherence, and second, consistency fine-tuning (ECT) to accelerate inference. Evaluated on the large-scale OGameData250K dataset, SoundReactor significantly outperforms the previous works in both audio quality and audio-visual alignment. Remarkably, it achieves real-time operation (≤ 32 ms per frame at 30 FPS), while maintaining comparable perceptual quality. Overall, SoundReactor establishes a new paradigm for causal, real-time V2A generation, providing a foundation for interactive, multimodal, and live content generation in the future.

**Strengths:**

1. This paper is well written and supported by extensive and systematic experiments.

2. This paper introduces a new frame-level online V2A generation paradigm. This enables real-time audio generation synchronized with streaming video frames, which is crucial for interactive applications such as gaming, virtual environments, and world models in robotics.

3. This paper achieves end-to-end causality from vision to audio, enabling frame-by-frame generation with minimal latency.

**Weaknesses:**

1. While the authors claim frame-level causality, the sampling can involve autoregressive decoding with a diffusion process, which may introduce temporal lookahead latency.

2.  Although “dinov2-vits14-reg” is used for efficiency, there’s no comparison to other vision encoders, except DINOv2's series. It’s unclear how much grid (patch) features extracted by DINOv2[1] actually contribute to performance.



[1] Oquab, Maxime, et al. “DINOv2: Learning robust visual features with-out supervision.” Transactions on Machine Learning Research, 2024
[2] Evans, Zach, et al. "Fast timing-conditioned latent audio diffusion." Forty-first International Conference on Machine Learning. 2024.
[3] Evans, Zach, et al. "Stable audio open." ICASSP 2025-2025 IEEE International Conference on Acoustics, Speech and Signal Processing (ICASSP). IEEE, 2025.

**Questions:**

1. During video token modeling, each frame’s features are concatenated with their temporal difference from the previous frame $(\hat{V}i - \hat{V}{i-1})$ to inject temporal cues. Could this differencing introduce a risk of future leakage? What would happen if you simply concatenated the previous frame features or injected sequence information using positional encoding instead?

2. Does the proposed framework ensure scalability for other-resolution videos and high-frequency audio (e.g., 96 kHz)?

3.How would the inference speed of the offline model change if it were limited to only the current and past frames instead of the full video sequence?

---

### Official Review · Reviewer_184U · 2025-11-01

**Soundness:** 3
**Presentation:** 3
**Contribution:** 3
**Rating:** 4
**Confidence:** 4

**Summary:**

This paper introduces a novel task—frame-level online video-to-audio (V2A) generation—where the model generates audio autoregressively from video frames without access to future frames, a setting critical for real-time applications like live content creation and interactive world models. To address this, the authors propose SoundReactor, the first framework explicitly designed for this task, featuring:
* A causal, decoder-only transformer operating on continuous audio latents.
* A novel visual conditioning scheme using DINOv2 grid features aggregated per frame with temporal differencing to ensure end-to-end causality and synchronization.
* A two-stage training pipeline: diffusion pre-training followed by consistency fine-tuning (via ECT) to accelerate inference.
Evaluated on the OGameData dataset, SoundReactor generates high-quality, full-band stereo audio with strong semantic and temporal alignment, achieving low per-frame token-level latency (26.6–30.3 ms on H100). Both objective metrics and human evaluations confirm its superiority over offline AR baselines like V-AURA, which violate causality. The model also demonstrates zero-shot generalization to longer sequences via NTK-aware RoPE extension. Ablation studies validate key design choices, including the diffusion head size, CFG strategy, and vision conditioning.

**Strengths:**

* Well-Designed Causal Framework: The proposed SoundReactor is the first framework explicitly designed for this challenging online setting, enforcing strict end-to-end causality—including causal vision encoding—which is essential for real-time operation without access to future frames.
* Effective and Efficient Architecture: The model leverages a lightweight DINOv2 encoder for efficient visual feature extraction, continuous audio latents via VAE for high-quality reconstruction, and a decoder-only transformer with a diffusion head, achieving a strong balance between audio quality and low per-frame latency.
* Strong Empirical Performance: Comprehensive experiments on a diverse gameplay dataset demonstrate that SoundReactor generates high-quality, full-band stereo audio with strong semantic and temporal alignment, outperforming strong offline AR baselines like V-AURA under the causal constraint.
* Practical Inference Acceleration: The integration of Easy Consistency Tuning (ECT) significantly reduces the number of function evaluations (NFE) in the diffusion head (down to 1–4 steps) while maintaining audio quality, making the model highly suitable for real-time applications.

**Weaknesses:**

* Limited Evaluation on Long-Form Sequence Generation: The method demonstrates frame-level generation and alignment capabilities, which should theoretically offer advantages over many existing diffusion-based models for long-sequence generation, as those often rely on concatenation of shorter segments. However, the current evaluation only extends to 16-second sequences, which is not particularly long. It would be valuable to test and compare the performance on significantly longer audio sequences to better validate the scalability and stability of the proposed online generation approach.
* Limited Generalizability to Real-World Datasets: While the model performs well on the curated OGameData dataset (comprising gameplay videos), its performance on the more diverse and uncurated VGGSound dataset is noticeably weaker, particularly in terms of semantic alignment (IB-Score) and audio quality (KL_PaSST). This suggests the model may not generalize as effectively to real-world, in-the-wild video content.
* Incomplete Inference Latency Comparison: To the best of our knowledge, other models also have implementations that focus on inference acceleration. It would be beneficial to supplement the experiments with comparative inference speed analyses against these accelerated baselines, providing a more comprehensive understanding of the practical efficiency advantages of the proposed method.

**Questions:**

Please refer to the weaknesses section.

**Details Of Ethics Concerns:**

No concerns.

---

### Note · Authors · 2025-11-13

I have read and agree with the venue's withdrawal policy on behalf of myself and my co-authors.